

# Measurement report: Three-year characteristics of sulfuric acid in urban Beijing and derivation of daytime sulfuric acid proxies applicable to various sites

Yishuo Guo[1,2], Chao Yan[2,3*], Chang Li[2], Chenjuan Deng[4], Ying Zhang[2,3], Ying Zhou[2], Haotian Zheng[4], Yueqi Jiang[4], Xin Chen[2], Wei Ma[2], Nina Sarnela[5], Zhuohui Lin[2], Chenjie Hua[2], Xiaolong Fan[6], Feixue Zheng[2], Zemin Feng[2], Zongcheng Wang[2], Yusheng Zhang[2], Jingkun Jiang[4], Bin Zhao[4], Markku Kulmala[2,3,5], Yongchun Liu[2*]

**Affiliations:**

[1] College of Environmental and Chemical Engineering, Hebei Vocational University of Industry and Technology, Shijiazhuang, Hebei, China

[2] Aerosol and Haze Laboratory, Beijing Advanced Innovation Center for Soft Matter Science and Engineering, Beijing University of Chemical Technology, Beijing, China

[3] Nanjing-Helsinki Institute in Atmospheric and Earth System Sciences, Nanjing University, Suzhou, China.

[4] State Key Joint Laboratory of Environment Simulation and Pollution Control, State Environmental Protection Key Laboratory of Sources and Control of Air Pollution Complex, School of Environment, Tsinghua University, Beijing 100084, China

[5] Institute for Atmospheric and Earth System Research / Physics, Faculty of Science, University of Helsinki, Finland

[6] Center for Excellence in Regional Atmospheric Environment, Key Lab of Urban Environment and Health, Institute of Urban Environment, Chinese Academy of Sciences, Xiamen, China

*Correspondence to: Chao Yan (chaoyan@nju.edu.cn), Yongchun Liu (liuyc@buct.edu.cn)*

**Abstract** Sulfuric acid ($H_2SO_4$) is a key precursor in atmospheric new particle formation and cluster early growth. However, long-term measurement of it remains scarce due to technical challenges. Although several proxies for estimating $H_2SO_4$ concentration have been proposed, they are always site-specific. Therefore, both reliable $H_2SO_4$ measurement and proxies with wider application are highly needed. Here, we conducted a long-term $H_2SO_4$ measurement in urban Beijing during 2019–2021, and derived three $H_2SO_4$ proxies based entirely on its formation and loss pathways (OH-CS, UVB-CS and UVB-$PM_{2.5}$ based proxies). Results show that daytime $H_2SO_4$ concentration is $2.0–7.4\times10^6$ molec cm$^{-3}$ and shows an overall decline with an average annual decrease of 14%. This decline is mainly due to the ongoing $SO_2$ emission controls. Daytime $H_2SO_4$ shows a clear seasonal variation that tracks UVB. Nighttime $H_2SO_4$ concentration is $1.6–6.3\times10^5$ molec cm$^{-3}$, with higher levels in warmer seasons due to stronger sources and lower condensation sink. The diurnal variations of $H_2SO_4$ across seasons follow those of photo-oxidation-related parameters, such as UVB, OH radical, and $J(NO_2)$. All of the three proxies can reproduce $H_2SO_4$ concentration during 10:00–14:00. Importantly, they can estimate $H_2SO_4$ concentration at a boreal forest site in Hyytiälä, Finland, suggesting their applicability to sites with diverse environments. Furthermore, the parameters used in UVB-$PM_{2.5}$ based proxy are available at most observational sites. Further application of this proxy could provide $H_2SO_4$ concentrations covering many regions worldwide, which may further facilitate research on atmospheric nucleation and secondary aerosol growth of these sites.

## 1. Introduction





New particle formation (NPF) is a key contributor to the born of atmospheric aerosols (Merikanto et al., 2009;Gordon et al., 2017), and thus can have a great influence on global climate and human health (Stocker, 2014;Lelieveld et al., 2015). Among all the precursors that drive atmospheric nucleation, the initial step of NPF, sulfuric acid ($H_2SO_4$) has been shown to be the most important one from both laboratory experiments and field observations (Kulmala et al., 2006;Riipinen et al., 2007;Paasonen et al., 2009;Erupe et al., 2010;Wang et al., 2011;Kirkby et al., 2011;Yu et al., 2012;Almeida et al., 2013;Kürten et al., 2014;Riccobono et al., 2014;Lehtipalo et al., 2018;Yao et al., 2018;Lee et al., 2019;Myllys et al., 2019;Yan et al., 2021). The clusters formed by sulfuric acid and base molecules, such as ammonia and amines, provide the primary core for further condensation of other low-volatility species, promoting aerosols growth to tens of nanometers, reaching the sizes of cloud condensation nuclei (CNN) and ultrafine particles. Therefore, reliable measurement of sulfuric acid is of great importance.

Since the 1990s, sulfuric acid measurements have been conducted in various field campaigns that covered a wide range of atmospheric environments, including urban (McMurry et al., 2005;Fiedler et al., 2005;Riipinen et al., 2007;Mikkonen et al., 2011;Wang et al., 2011;Yao et al., 2018;Lu et al., 2019), rural (Berresheim et al., 2000;Birmili et al., 2003;Paasonen et al., 2009;Erupe et al., 2010;Mikkonen et al., 2011;Kürten et al., 2016;Yang et al., 2021a;Yang et al., 2023), mountainous (Weber et al., 1996;Weber et al., 1997;Boy et al., 2008;Mikkonen et al., 2011), marine (Berresheim et al., 1993;Weber et al., 1995;Weber et al., 1996;Berresheim et al., 2002;O'Dowd et al., 2002), forest environments (Fiedler et al., 2005;Riipinen et al., 2007;Petäjä et al., 2009;Nieminen et al., 2009;Mikkonen et al., 2011;Jokinen et al., 2012), among others (Weber et al., 1998;Mauldin et al., 2001;Mauldin et al., 2004;Sarnela et al., 2015;Jokinen et al., 2018). The locations of these sites, measurement periods, and corresponding sulfuric acid concentrations are summarized in Table 1. In general, within the planetary boundary layer, sulfuric acid concentration was around $0.2 - 15 \times 10^6$ cm$^{-3}$, with the highest levels in urban areas, followed by rural, mountainous, and marine regions, and the lowest in forest areas. This suggests that sulfuric acid levels depend strongly on the intensity of human activity. In addition, most measurement campaigns lasted for less than four months and concentrated mostly on warmer seasons (spring, summer and early autumn) when NPF usually occurs (Dal Maso et al., 2005;Manninen et al., 2009;Dada et al., 2017;Nieminen et al., 2018;Chu et al., 2019;Qi et al., 2015). Previous studies showed that NPF in Chinese megacities was also frequently observed in winter (Deng et al., 2020;Chu et al., 2019), and thus sulfuric acid measurement in cold seasons is also crucial. To date, however, long-term measurement of sulfuric acid worldwide is still lacking, which somewhat limits the investigation of NPF processes.

To complement the limited sulfuric acid measurement, several sulfuric acid proxies were developed. The original expression for estimating sulfuric acid concentration was derived from its production and loss pathways. Assuming that sulfuric acid originates solely from OH-initiated oxidation of $SO_2$ and the only loss is the condensation sink (CS) onto particle surfaces, the steady-state concentration of sulfuric acid can be expressed as $[H_2SO_4] = k \cdot [OH][SO_2]/CS$, where $k$ is the rate constant of OH + $SO_2$ reaction. As early as 1997, Weber et al. estimated sulfuric acid concentrations at a marine site and a mountain site using this expression (Weber et al., 1996). Results showed





that the estimated daytime sulfuric acid for two selected days generally matched the measured one at both sites.
Later, Berresheim et al. also utilized this expression at a coastal site (Berresheim et al., 2002). However, when the
accommodation coefficient of CS calculation was chosen as 1, the estimated sulfuric acid concentration turned out
**Table 1.** Summary of atmospheric sites with sulfuric acid measurement.

| Type of Site | Location | Measurement Period | $H_2SO_4$ ($\times 10^6$ molec cm$^{-3}$) | References |
|---|---|---|---|---|
| Urban | Atlanta, Georgia, USA | 2002/08 | 2.9 | 2005, McMurry et al. 2011, Mikkonen et al. |
| | Heidelberg, Germany | 2004/03 - 04 | 3.0 | 2005, Fiedler, et al. 2007, Riipinen et al. |
| | Beijing, China | 2008/07 - 09 | 1.0 - 9.0 | 2011, Wang et al. |
| | Shanghai, China | 2014/03 - 2016/02 | 7.8 | 2018, Yao et al. |
| | Beijing, China | 2018/02 - 03 | 4.9 | 2019, Lu, et al. |
| Rural | Hohenpeissenberg, Germany | 151 days in 1998 186 days in 1999 | 0.1 -10 | 2000, Berresheim et al. |
| | | 1998/04 - 2000/07 | 0.6 | 2003, Birmili et al. 2009, Paasonen et al. 2011, Mikkonen et al. |
| | Ohio, USA | 2008/08 - 2009/11 | winter: 0.6 spring: 5.2 summer: 2.9 autumn: 0.5 | 2010, Erupe et al. |
| | San Pietro Capofiume, Italy | 2009/06 - 07 | 2.4 | 2011, Mikkonen et al. |
| | Melpitz, Germany | 2008/05 | 2.9 | 2011, Mikkonen et al. |
| | Vielbrunn, Germany | 2014/05 - 06 | 3.0 | 2016, Kürten et al. |
| | Nanjing, China | 2017/12 - 2018/01 2018/04 2018/07 - 08 2018/11 | winter: 1.9 spring: 7.4 summer: 4.5 autumn: 9.0 | 2021, Yang et al. |
| | Xiamen, China | 2022/07 - 08 | 2.3 | 2023, Yang et al. |
| Mountain | Colorado, USA | 1993/09 | 0.1 - 10 | 1996, Weber et al. 1997, Weber et al. |
| | | 2006/06 - 07 | 2.8 | 2008, Boy et al. |
| | | 2007/06 - 07 | 1.4 | 2011, Mikkonen et al. |
| Marine | Washington, USA | 1991/04 | 0.03 - 32.0 | 1993, Berresheim et al. |
| | Hawaii, USA | 1992/07 | upslope: 1.2 downslope: 0.5 | 1995, Weber et al. 1996, Weber et al. |
| | Mace Head, Ireland | 1999/06 | 1.5 | 2002, Berresheim et al. |
| | | 1998/09 1999/06 | 2.0 - 15.0 | 2002, O'Wowd et al. |
| Forest | Hyytiälä, Finland | 2003/03 - 04 | 2.6 | 2005, Fiedler et al. |
| | | 2005/04 - 05 | 1.0 - 10 | 2007, Riipinen et al. |
| | | 2007/03 - 06 | 0.9 - 2.5 | 2009, Petäjä et al. 2009, Nieminen et al. |
| | | 2003/03 - 04 | 0.6 | 2011, Mikkonen et al. |
| | | 2007/03 - 06 | 0.2 | 2011, Mikkonen et al. |
| | | 2011/03 - 04 | 0.3 - 10 | 2012, Jokinen et al. |
| Others | Macquarie Island, Australia | 11/27/1995 | 2.8 | 1998, Weber et al. |
| | Antarctica | 1998/12 | 0.3 | 2001, Mauldin III et al. |
| | | 2000/11 - 12 | 0.3 | 2004, Mauldin III et al. |
| | | 2014/11 - 2015/01 | 0.2 - 10 | 2018, Jokinen et al. |
| | Kilpilahti, Finland | 2012/06 - 07 | oil refinery: 11.5 industrial: 4.4 non-industrial: 1.3 | 2015, Sarnela et al. |



to be much lower than the measured one. The authors speculated that additional sulfuric acid sources might exist,
likely the OH- or BrO-initiated oxidation of dimethyl disulfide or dimethyl sulfide, or the oxidation of $SO_2$ by non-
OH oxidants. This also suggests that this proxy may not be suitable for coastal environments. In 2009, Petäjä et al.
proposed the concept of sulfuric acid proxy clearly and derived three proxies based on its source-sink equilibrium
(Petäjä et al., 2009). The first proxy, $P_1 = k_1 \cdot [OH][SO_2]/CS$, was very similar to that proposed by Weber et al., but
the pre-factor $k_1$ was obtained by the fitting of measurement data. During daytime, OH radical mainly arises from
photochemical reactions. Therefore, OH radical in $P_1$ could be replaced by UVB, yielding the second proxy of $P_2 =$
$k_2 \cdot [UVB][SO_2]/CS$. Similarly, replacing OH radical with global radiation yielded the third proxy, $P_3 =$
$k_3 \cdot [Glob][SO_2]/CS$. These three proxies showed good performance in estimating daytime sulfuric acid concentration.
However, the authors noted that $k_1$, $k_2$ and $k_3$ came from fits so that they are likely site-specific, which limits their
transferability to other sites.
Later, Mikkonen et al. attempted to develop sulfuric acid proxies suitable for various environments. The datasets
came from five sites, including one forest, one mountainous, two rural and one urban sites (Mikkonen et al., 2011).
Five linear-fitting proxies, including one ($L1 = B \cdot k \cdot Radiation \cdot [SO_2] \cdot CS^{-1}$) similar to the $P_3$ proxy proposed by Petäjä
et al., were first built. Results showed only minor differences among the five proxies, with L3 proxy ($L3 =$
$B \cdot k \cdot Radiation \cdot [SO_2]^{0.5}$) generally performing the best. Based on this, the authors concluded that the pseudo-steady-
state assumption for gaseous sulfuric acid could be somewhat unrealistic in atmospheric conditions, and then
proposed five additional nonlinear-fitting sulfuric acid proxies. However, only correlation coefficients were used to
evaluate the performance of five linear fitting proxies, and these correlation coefficients were close to each other.
Thus, the above conclusion requires more data to be supported.
Based on these studies, Lu et al. developed seven nonlinear-fitting proxies to estimate daytime sulfuric acid
concentration in urban Beijing (Lu et al., 2019). In proxies of N5–N7, $O_3$ and HONO were included to account for
OH radical formation via photolysis of HONO. Results showed that seven proxies generally performed well in
estimating daytime sulfuric acid concentration with similar correlation coefficients and relative errors. Nevertheless,
the authors concluded that N7 proxy ($[H_2SO_4] = 0.0013 \cdot UVB^{0.13} \cdot [SO_2]^{0.40} \cdot CS^{-0.17} \cdot ([O_3]^{0.44}+[NO_x]^{0.41})$) was the most
suitable for estimating daytime sulfuric acid, as it took the CS loss pathway into account, had the lowest relative
error and used the easily measured $NO_x$. Note that this proxy was developed for urban Beijing, and thus may not
apply to other sites. A year later, Dada et al. constructed proxies based on the source-sink equilibrium of sulfuric
acid at four different sites, including one boreal forest, one rural, one urban, and one megacity sites. The formation
of sulfuric acid from the ozonolysis of alkenes was first considered. Results showed that $P_1$ ($[H_2SO_4] = -$
$\frac{CS}{2k_3} + \sqrt{\left(\frac{CS}{2k_3}\right)^2 + \frac{[SO_2]}{k_3}(k_1 \cdot GlobRad + k_2 \cdot [O_3][Alkenes])}$) and $P_3$ ($[H_2SO_4] = \frac{k_1 \cdot GlobRad[SO_2] + k_2 \cdot [SO_2][O_3][Alkene]}{CS}$) proxies
with the alkene ozonolysis term could estimate both daytime and nighttime sulfuric acid well, while $P_2$ proxy
($[H_2SO_4] = -\frac{CS}{2k_3} + \sqrt{\left(\frac{CS}{2k_3}\right)^2 + \frac{[SO_2]}{k_3}k_1 \cdot GlobRad}$) without the alkene ozonolysis term could only estimate daytime



sulfuric acid. Although the proxy equations were the same across sites, the parameters therein were different. Thus,
these four proxies have limited application at other sites.
In this study, we characterize the interannual, seasonal, and diurnal variations of sulfuric acid in urban Beijing
and derive proxies to estimate sulfuric acid concentration at various sites. Long-term measurement of sulfuric acid
covering nearly three continuous years (from 1$^{st}$ January, 2019 to 11$^{th}$ November, 2021) was conducted in urban
Beijing. First, the yearly and seasonal variation of sulfuric acid concentration, as well as its diurnal cycles were
analyzed. Second, the performance of nine representative proxies, including seven steady-state based ones and two
numerical regression ones, from previous studies at our site was investigated. Based on these analyses, three steady-
state proxies were proposed according to the budget analysis of sulfuric acid, and their performance and limitations
on estimating daytime sulfuric acid concentration were investigated in detail. These three proxies were then applied
to estimate sulfuric acid concentration at a boreal forest site in Hyytiälä, Finland. Correlation coefficients and
relative errors indicate that three proxies are able to reproduce daytime sulfuric acid well, suggesting that three
proxies and parameters therein could be applicable at other atmospheric sites. Finally, a general suggestion on proxy
selection with different available parameters was given.
**2. Method**
**2.1 Measurement site**
The measurements were conducted at the Aerosol and Haze Laboratory at the west campus of Beijing University
of Chemical Technology (39.95° N, 116.31° E). It is a typical urban site surrounded by commercial and residential
areas and three major roads (Liu et al., 2020;Yan et al., 2021;Guo et al., 2021;Yan et al., 2022). The datasets used
in this study span nearly three continuous years from January 2019 to November 2022.
**2.2 Measurement of sulfuric acid**
Sulfuric acid was measured by a long time-of-flight chemical ionization mass specter (LTOF-CIMS, Aerodyne
Research, Inc.) using nitric acid as reagent ions. The basic working principle of this instrument is described
elsewhere (Jokinen et al., 2012), and the instrument configuration has been provided in our previous studies and
has remained unchanged over the years. Briefly, air was drawn through a stainless-steel tube (1.6 m long, 3/4 inch
in diameter). The inlet flow rate was maintained at 7.2 L min$^{-1}$. Additionally, a flush plate (Karsa Inc.) was installed
to effectively remove water vapor in the sampled air.
Sulfuric acid concentration was quantified from the ratio of bisulfate ions (with counting rates unit in ions s$^{-1}$)
to primary ions as follows:
$$H_2SO_4 = \frac{HSO_4^- + H_2SO_4NO_3^-}{NO_3^- + HNO_3NO_3^- + (HNO_3)_2NO_3^-} \times C$$

where C is the calibration coefficient, determined by direct calibration using known amounts of gaseous sulfuric
acid injected into the instrument (Kürten et al., 2012). During the measurement period, the instrument ran stably.
Calibration was performed every six months and after tuning. After correcting the diffusional wall loss of the



sampling line (0.2129), the final calibration coefficients were 6.07 - 7.47 × 10$^9$ molec cm$^{-3}$ over the three-year
period.

**2.3 Other ancillary measurements**

Particle number concentration and size distribution was measured by a differential mobility particle sizer (DMPS,
6 – 840 nm) (Aalto et al., 2001) and a particle size distribution system (PSD, 3 nm – 10 µm) (Liu et al., 2016). The
configuration of these two instruments have been described in our previous studies (Zhou et al., 2021, Yan et al.,
2021). Based on these measurements, the condensation sink (CS) of sulfuric acid can be calculated from the
following equation (Kulmala et al., 2012):

$$CS = 4\pi D \int_0^{d_p max} \beta_m(d_p')d_p'N_{d_p'}\,dd_p' = 4\pi D \sum_{d_p'} \beta_{m,d_p'}d_p'N_{d_p'}$$

where $D$ is the diffusion coefficient of sulfuric acid, $d_p'$ is the particle diameter, $N_{d_p'}$ is the particle number
concentration with diameter $d_p'$, and $\beta_m$ represents the transition-regime correction. Size-resolved hygroscopic
growth of aerosols was considered in the calculation of CS. CS values calculated from two instruments are shown
in Figure S13. The datasets from PSD was chosen in priority as it measures wider size ranges. If PSD data was
unavailable or not consecutive for more than 10–20 days, DMPS data was used. There were three periods during
which measurements from both instruments were continuous and stable, and the CS comparison for these periods
are shown in Figure S14. Compared with PSD CS values, DMPS CS values were on average ~ 11.7% lower.
Meteorological parameters were measured by a weather station (AWS310, Vaisala Inc.) located on the building
rooftop. These parameters include ambient temperature, relative humidity (RH), pressure, visibility, UVB radiation,
and horizontal wind speed and direction. Trace gases, including carbon monoxide (CO), sulfur dioxide (SO$_2$),
nitrogen oxides (NO$_x$), and ozone (O$_3$), were monitored using four Thermo Environmental Instruments (models 48i,
43i-TLE, 42i, 49i, respectively). Calibrations of these instruments were performed every two weeks using standard
gases of known concentrations. The mass concentration of PM$_{2.5}$ and PM$_{10}$ were measured with a tapered element
oscillating microbalance dichotomous ambient particulate monitor (TEOM 1405-DF, Thermo Fisher Scientific Inc,
USA). The mass concentration of PM$_{coarse}$ was obtained based on the difference between PM$_{10}$ and PM$_{2.5}$.

**2.4 Modelling of OH radical, J(NO$_2$) and J(O$^1$D)**

The Weather Research and Forecasting Model-Community Multiscale Air Quality (WRF-CMAQ) model was
applied to simulate the OH radical concentration, J(NO$_2$) and J(O$^1$D). The simulation period covered the entire year
of 2019 and 1$^{st}$ January to 19$^{th}$ February, 2020. The physical options in WRF (version 3.9.1) were the same as in
Zheng et al. (Zheng et al., 2019) The CMAQ model (version 5.3.2) couple with the two-dimensional Volatility
Basis Set (2D-VBS) (Zhao et al., 2016) adopted SAPRC07 mechanism was used for gas-phase chemistry, and the
AERO6 (Sarwar et al., 2011) was used for aerosol module. The horizontal resolution was 27 km × 27 km and the
vertical extent was divided into 14 layers. The modelling domain was the same as in Zheng et al. (Zheng et al.,

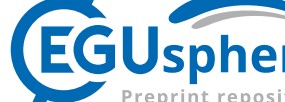

2020). To minimize the influence of initial conditions, the modelling simulations started 5 days before the modelling
period. Default boundary layer settings were used in the simulations.

## 3. Results and discussion

### 3.1 Characteristics of measured sulfuric acid

#### 3.1.1 Interannual and monthly variations of sulfuric acid

In the early morning, sulfuric acid concentration is influenced not only by the photochemical production and
loss pathways, but also by additional sources, such as $SO_2$ oxidation on traffic-related black carbon (Yao et al.,
2020). In addition, sulfuric acid from direct emission and the ozonolysis of alkenes cannot be ignored during
daytime when far from noon (Yang et al., 2021a). Therefore, daytime window of 10:00–14:00 (local time) was
chosen for proxy evaluation unless specified otherwise. This period also corresponds to the new particle formation
time (Kulmala et al., 2007;Kulmala et al., 2013;Deng et al., 2020;Ma et al., 2021). The corresponding nighttime
window was 22:00–02:00 next day (local time).

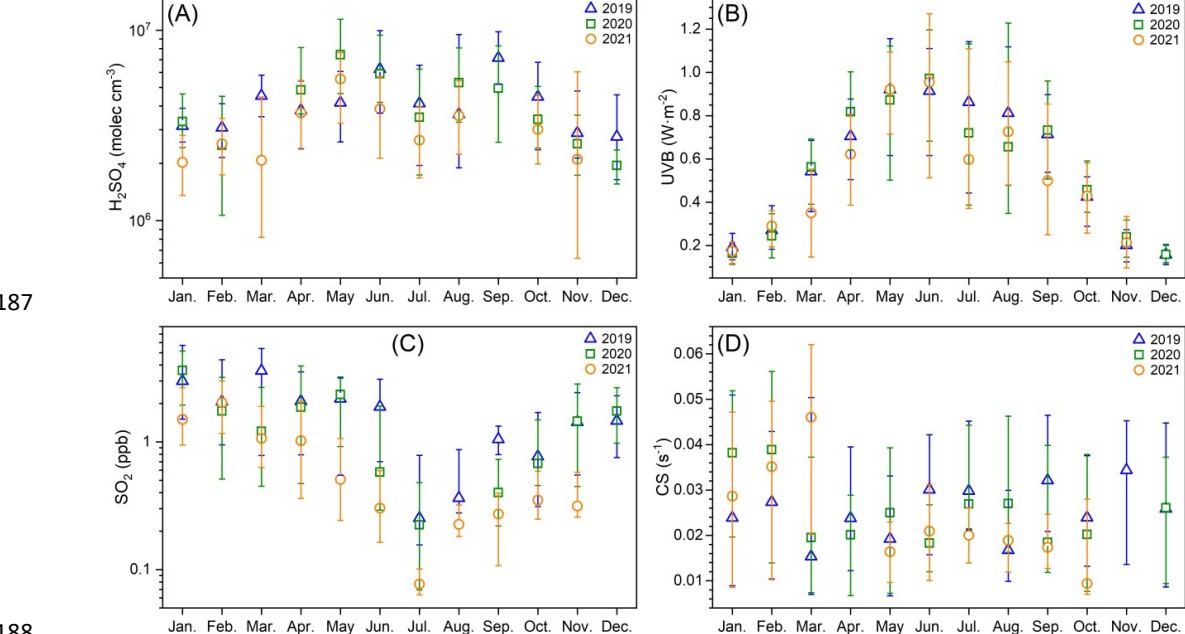

**Figure 1.** Three-year (from 2019 January to 2021 November) monthly variations of (A) $H_2SO_4$ concentration, (B) UVB, (C)
$SO_2$ and (D) condensation sink (CS) during daytime (10:00-14:00) when new particle formation mostly occurs. Blue triangles,
green squares and orange circles represent data in 2019, 2020 and 2021, respectively. The up line, middle marker and bottom
line stand for upper quartile, median and lower quartile values, respectively.

In urban Beijing, typical daytime sulfuric acid concentration ranges from $2.0 \times 10^6$ to $7.4 \times 10^6$ molec cm$^{-3}$
(monthly median concentration, Figure 1A and Table 2). Figure 1A and Figure S1E show that sulfuric acid
concentration generally declines from 2019 to 2021, with an average annual decrease of 14%. During the three
years, UVB intensity remains roughly constant (Figure 1B and Figure S1A), CS change is not significant (Figure
1D and Figure S1E), while $SO_2$ concentration decreases markedly (Figure 1C and Figure S1D). Thus, the yearly



decline of sulfuric acid is mainly attributed to the decrease of $SO_2$ (by ~25% per year). Figure 1A also reveals that
sulfuric acid concentration has a clear seasonal variation, which is the highest in May ($4.2 - 7.4 \times 10^6$ molec cm$^{-3}$)
and September ($5.0 - 7.2 \times 10^6$ molec cm$^{-3}$) and the lowest from November to February of the next year ($2.0 - 3.3$
$\times 10^6$ molec cm$^{-3}$). UVB shows the same monthly pattern as sulfuric acid, which reaches the highest from May to
September and decreases to the lowest from November to February of next year, while $SO_2$ shows an opposite
monthly trend to sulfuric acid. This indicates that the influence of UVB on sulfuric acid monthly variation
outperforms that of $SO_2$. Meanwhile, sulfuric acid concentration in July is much lower from May to September,
likely driven by extremely low $SO_2$ despite a small decrease of UVB in that month. In Beijing, precipitation occurs
more frequently in July and August than in May, June and September (Table S1). This reduces UVB and $SO_2$ in
these two months (Figure S2), and further leads to lower sulfuric acid concentration. Overall, UVB intensity and
$SO_2$ concentration are the two key parameters determining sulfuric acid concentration.
**Table 2.** Monthly concentration of $H_2SO_4$ (molec cm$^{-3}$) during daytime (10:00-14:00) from 2019 to 2021. "NaN" means there
is no data available.

| Month | 2019 | | | 2020 | | | 2021 | | |
|---|---|---|---|---|---|---|---|---|---|
| | Median | 25th | 75th | Median | 25th | 75th | Median | 25th | 75th |
| January | $3.1\times10^6$ | $2.6\times10^6$ | $3.9\times10^6$ | $3.3\times10^6$ | $2.4\times10^6$ | $4.6\times10^6$ | **$2.0\times10^6$** | $1.4\times10^6$ | $2.8\times10^6$ |
| February | $3.1\times10^6$ | $2.1\times10^6$ | $4.1\times10^6$ | $2.5\times10^6$ | $1.1\times10^6$ | $4.5\times10^6$ | $2.5\times10^6$ | $1.7\times10^6$ | $3.4\times10^6$ |
| March | $4.5\times10^6$ | $3.5\times10^6$ | $5.8\times10^6$ | NaN | NaN | NaN | $2.1\times10^6$ | $8.2\times10^5$ | $4.5\times10^6$ |
| April | $3.8\times10^6$ | $2.4\times10^6$ | $5.4\times10^6$ | $4.9\times10^6$ | $3.6\times10^6$ | $8.1\times10^6$ | $3.7\times10^6$ | $2.4\times10^6$ | $5.4\times10^6$ |
| May | $4.2\times10^6$ | $2.6\times10^6$ | $6.1\times10^6$ | **$7.4\times10^6$** | $4.7\times10^6$ | $1.1\times10^7$ | $5.5\times10^6$ | $3.3\times10^6$ | $7.6\times10^6$ |
| June | $6.2\times10^6$ | $3.7\times10^6$ | $1.0\times10^7$ | $5.9\times10^6$ | $4.2\times10^6$ | $9.4\times10^6$ | $3.9\times10^6$ | $2.1\times10^6$ | $5.7\times10^6$ |
| July | $4.1\times10^6$ | $2.0\times10^6$ | $6.6\times10^6$ | $3.5\times10^6$ | $1.7\times10^6$ | $6.3\times10^6$ | $2.6\times10^6$ | $1.7\times10^6$ | $4.0\times10^6$ |
| August | $3.6\times10^6$ | $1.9\times10^6$ | $9.5\times10^6$ | $5.3\times10^6$ | $3.3\times10^6$ | $8.1\times10^6$ | $3.6\times10^6$ | $2.2\times10^6$ | $5.4\times10^6$ |
| September | $7.2\times10^6$ | $5.2\times10^6$ | $9.8\times10^6$ | $5.0\times10^6$ | $2.6\times10^6$ | $8.3\times10^6$ | NaN | NaN | NaN |
| October | $4.5\times10^6$ | $2.4\times10^6$ | $6.8\times10^6$ | $3.4\times10^6$ | $2.4\times10^6$ | $5.1\times10^6$ | $3.0\times10^6$ | $2.0\times10^6$ | $4.5\times10^6$ |
| November | $2.9\times10^6$ | $2.1\times10^6$ | $4.8\times10^6$ | $2.5\times10^6$ | $1.7\times10^6$ | $3.6\times10^6$ | $2.1\times10^6$ | $6.3\times10^5$ | $6.0\times10^6$ |
| December | $2.8\times10^6$ | $1.6\times10^6$ | $4.6\times10^6$ | **$2.0\times10^6$** | $1.6\times10^6$ | $2.4\times10^6$ | NaN | NaN | NaN |

Typical nighttime sulfuric acid concentration of urban Beijing ranges from $1.6 \times 10^5$ to $6.3 \times 10^5$ molec cm$^{-3}$
(monthly median concentration, Figure S3 and Table S2), about one order of magnitude lower than that of daytime.
Unlike daytime sulfuric acid, nighttime sulfuric acid concentration does not show a decreasing trend from 2019 to
2021. At night, under clean conditions, alkene ozonolysis is a major source of sulfuric acid (Guo et al., 2021); under
more polluted conditions, primary emissions from vehicles or fresh plumes indicated by benzene also play an
important role (Yang et al., 2021a). However, data of alkenes and benzene is unavailable in July–August 2019 and
in 2020–2021, making it impossible to estimate the intensities of these nocturnal sulfuric acid sources. Thus, we are
not able to give further explanation on the yearly variation of nighttime sulfuric acid. Figure S3A shows that
nighttime sulfuric acid has a similar but weaker seasonal variation as that of daytime, i.e., concentration is generally
higher from May to September than in other months, and concentration in July and August is significantly lower
than in May, June and September. According to the data of 2019, the direct-emission source is higher from March
to June (Figure S3B), and alkene-ozonolysis source is higher from March to September (Figure S3C), indicating



that the sources of nighttime sulfuric acid are stronger in warmer seasons. Meanwhile, the CS level is lower from
April to October (Figure S3D), resulting in lower losses of sulfuric acid. Together, these two factors lead to higher
nighttime sulfuric acid concentrations during warmer seasons.

### 3.1.2 Diurnal variations of sulfuric acid and related parameters

The diurnal patterns of sulfuric acid across seasons are similar, starting increasing in the early morning ($\sim$ 04:00–
06:00), peaking around noon ($\sim$ 11:00), and decreasing to a low level at nightfall ($\sim$19:00) (Figure 2A). The morning
increase of sulfuric acid occurs earliest in summer, followed by spring, autumn and winter. Moreover, the peak
width of sulfuric acid from the widest to narrowest follows the same seasonal trend. These diurnal patterns across
the four seasons resemble those of UVB (Figure 2B), suggesting that radiation-driven photochemical reactions
govern sulfuric acid formation. Moreover, the morning increase of sulfuric acid occurs earlier than UVB and global
radiation but close to $J(NO_2)$ and OH radical (Figure S4D). This suggests that UVB and global radiation are not
able to adequately represent the photochemical sources in early morning, whereas $SO_2$ oxidation by OH radicals
produced by $NO_2$ photolysis is a major source. HONO photolysis is another major formation pathway for OH radical
(Tan et al., 2017;Tan et al., 2018;Ma et al., 2019;Yang et al., 2021b;Ma et al., 2022). HONO decreases in the
morning at $\sim$ 06:00–07:00, more than an hour after the morning increase of sulfuric acid and OH radical (Figure
S4D). This suggests that sulfuric acid formation in the early morning is not likely caused by the oxidation of OH
radical from HONO photolysis. The daytime peaking hour of sulfuric acid is close to $J(NO_2)$ and $J(O^1D)$ (Figure
S4B), indicating that sulfuric acid peaking hour is controlled by photochemical reactions related to $J(NO_2)$ and
$J(O^1D)$. The peak width of sulfuric acid is the widest, followed by $J(NO_2)$, global radiation, OH radical, $J(O^1D)$ and
UVB (Figure S4C). This implies that when using proxies with these parameters to estimate daytime sulfuric acid
concentration, differences in peaking hours and peak widths may cause deviations from the measured concentration.
During the day, sulfuric acid concentration is the highest in summer, followed by spring and autumn, and then
winter. This seasonal variation generally tracks UVB, except in autumn, when UVB and $SO_2$ are lower than spring,
but daytime sulfuric acid concentration remains comparable to that in spring. In autumn, the frequency of sulfuric
acid with high concentrations ($\geq 1.2 \times 10^7$ molec cm$^{-3}$) is higher than spring (Figure S5), likely contributing to the
overall higher level of sulfuric acid. At night, sulfuric acid concentrations are comparable across seasons, even
though $SO_2$ and CS levels vary. As aforementioned, additional sources such as benzene-related emissions (Yang et
al., 2021a) are among the main nighttime sources of sulfuric acid, so nighttime sulfuric acid cannot be easily
interpreted by proxies only including $SO_2$, CS and OH radical.





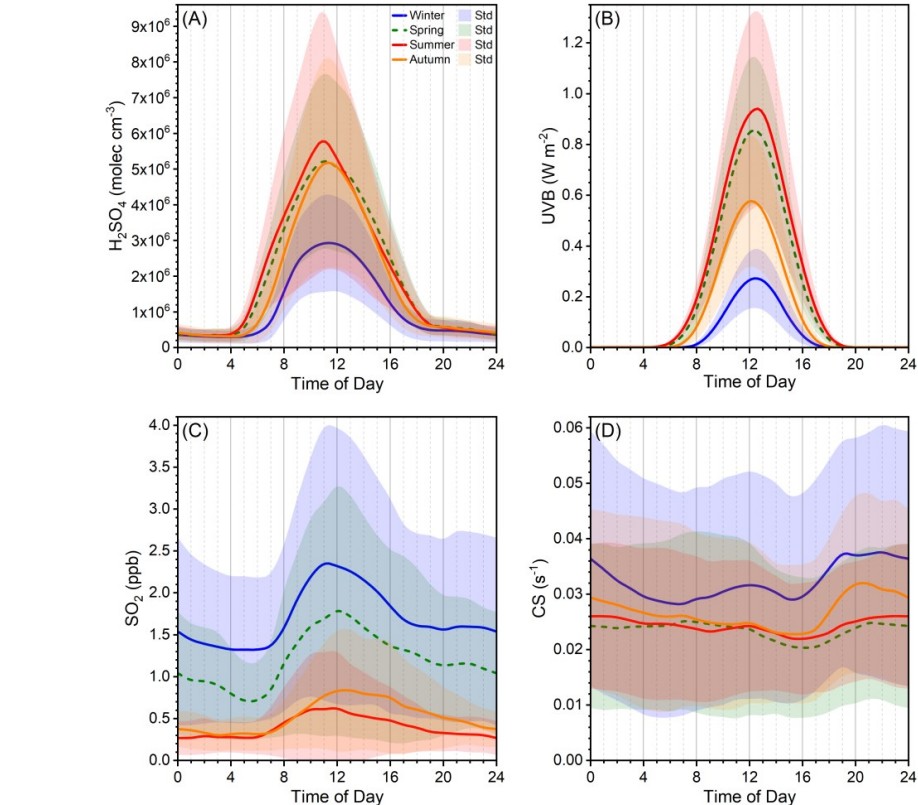

**Figure 2.** Three-year (from 2019 to 2021) diurnal variations of (A) $H_2SO_4$, (B) UVB, (C) $SO_2$ and (D) condensation sink (CS). Winter, spring, summer and autumn periods cover 15th November to 15th March of next year, 16th March to May, June to August, and September to 14th November, respectively. Lines are the mean values, and shaded areas denote the standard deviations of the data.

### 3.2 Performance of sulfuric acid proxies from previous study

In previous studies, several representative proxies for sulfuric acid that incorporate real physical and chemical considerations have been proposed. Before constructing the proxy for this study, we first evaluate the performance of existing proxies on estimating sulfuric acid concentration. The equations and internal parameters of these nine proxies are listed in Table 3 (Petäjä et al., 2009;Mikkonen et al., 2011;Lu et al., 2019;Dada et al., 2020). Figure 3 shows the measured sulfuric acid concentration and the proxy-estimated concentrations during daytime (10:00–14:00) in 2019. Table S3 summarizes the corresponding mean, standard deviation, median, lower quartile and upper quartile of sulfuric acid concentrations. To have a more accurate understanding on proxy performance, the correlation coefficients, power exponents, and slopes of the linear fittings between measured and estimated sulfuric acid, as well as the relative errors of estimated to measured sulfuric acid concentrations are further evaluated (Table 4). The relative error is calculated as follows (Lu et al., 2019):

$$\text{RE} = \frac{1}{n} \cdot \sum_{i=1}^{n} \frac{|[H_2SO_4]_{pro,i} - [H_2SO_4]_{mea,i}|}{[H_2SO_4]_{mea,i}}$$





**Table 3.** The equations and internal parameters of nine sulfuric acid proxies from literatures.

| Proxy | Equation | Parameters | Reference |
|---|---|---|---|
| $Proxy_{Petäjä\ OH-C}$ | $P_1 = \dfrac{k_1 \cdot [SO_2] \cdot [OH]}{CS}$ | $k_1 = 2.2 \times 10^{-12}$ cm³ molec⁻¹ s⁻¹ | (Petäjä et al., 2009) |
| $Proxy_{Petäjä\ OH-F}$ | | $k_1 = 8.6 \times 10^{-10} \times [OH]^{-0.48}$ cm³ s⁻¹ | |
| $Proxy_{Petäjä\ UVB-C}$ | $P_2 = \dfrac{k_2 \cdot [SO_2] \cdot UVB}{CS}$ | $k_2 = 9.9 \times 10^{-7}$ m² W⁻¹ s⁻¹ | |
| $Proxy_{Petäjä\ UVB-F}$ | | $k_2 = 8.4 \times 10^{-7} \times UVB^{-0.68}$ m² W⁻¹ s⁻¹ | |
| $Proxy_{Petäjä\ Glob-C}$ | $P_3 = \dfrac{k_3 \cdot [SO_2] \cdot Glob}{CS}$ | $k_3 = 2.3 \times 10^{-9}$ m² W⁻¹ s⁻¹ | |
| $Proxy_{Petäjä\ Glob-F}$ | | $k_3 = 1.4 \times 10^{-7} \times Glob^{-0.70}$ m² W⁻¹ s⁻¹ | |
| $Proxy_{Mikkonen\ et\ al.}$ | $a \cdot k \cdot Radiation^b \cdot [SO_2]^c \cdot (CS \cdot RH)^f$ | $a = 8.21 \times 10^{-3}$, $b=1$, $c=0.62$, $f=-0.13$ | (Mikkonen et al., 2011) |
| $Proxy_{Lu\ et\ al.}$ | $k_0 \cdot UVB^a \cdot [SO_2]^b \cdot CS^c \cdot (O_3{}^d + NO_x{}^e)$ | $k_0 = 0.0013$, $a=0.13$, $b=0.40$, $c=-0.17$, $e=0.44$, $f=0.41$ | (Lu et al., 2019) |
| $Proxy_{Lubna\ et\ al.}$ | $-\dfrac{CS}{2k_3} + \sqrt{\left(\dfrac{CS}{2k_3}\right)^2 + \dfrac{[SO_2]}{k_3} k_1 Glob}$ | $k_1 = 1.0 \times 10^{-6}$, $k_3 = 1.6 \times 10^{-7}$ | (Dada et al., 2020) |

As shown in Figure 3 and Table 4, estimated sulfuric acid concentrations from $Proxy_{Petäjä\ OH-C}$, $Proxy_{Petäjä\ OH-F}$
and $Proxy_{Lu\ et\ al.}$ are closest to the measured one, with median deviations within twofold. This suggests that these
three proxies provide the best estimates of sulfuric acid concentration. Estimated sulfuric acid concentrations from
$Proxy_{Petäjä\ Glob-C}$ and $Proxy_{Dada\ et\ al.}$ are not too far away from the measured one, with median deviations within
threefold. While estimated concentrations from other proxies differ substantially from the measured one, especially
$Proxy_{Petäjä\ UVB-F}$, which underestimates sulfuric acid concentration markedly. The linear correlation coefficients of
$Proxy_{Petäjä\ OH-C}$, $Proxy_{Petäjä\ OH-F}$ and $Proxy_{Petäjä\ UVB-C}$ proxies are closet to 1.0. Similarly, the power exponents of
$Proxy_{Petäjä\ OH-C}$, $Proxy_{Petäjä\ OH-F}$, $Proxy_{Petäjä\ UVB-C}$ and $Proxy_{Petäjä\ Glob-C}$ are closet to 1.0. This indicates that the estimated
sulfuric acid from former three proxies have the best linear correlation with the measurement. Only the slope of
$Proxy_{Petäjä\ OH-C}$ (0.85) is close to 1.0, suggesting that it performs the best in linear relationship. For $Proxy_{Petäjä\ UVB-F}$,
$Proxy_{Petäjä\ Glob-F}$, $Proxy_{Mikkonen\ et\ al.}$, $Proxy_{Lu\ et\ al.}$ and $Proxy_{Dada\ et\ al.}$, none of the linear correlation coefficients, power
exponents or the slopes perform well, indicating that they fail to reproduce the linearity with measured sulfuric acid.
The relative errors are within 50% for $Proxy_{Petäjä\ OH-F}$, $Proxy_{Petäjä\ Glob-C}$ and $Proxy_{Petäjä\ Glob-F}$, and those for $Proxy_{Petäjä}$
$_{OH-C}$, $Proxy_{Petäjä\ UVB-C}$, $Proxy_{Petäjä\ UVB-F}$ and $Proxy_{Lu\ et\ al.}$ range from 57% to 91%.
Considering both linear correlation and concentration estimation accuracy, $Proxy_{Petäjä\ OH-C}$ and $Proxy_{Petäjä\ OH-F}$ are
the two most suitable proxies for reproducing sulfuric acid concentration. Other four proxies from Petäjä et al., 2009
without the OH radical term underestimate the concentration of sulfuric acid. The reason might be that the scaling
factors $k_2$ and $k_3$ were obtained by fitting measured sulfuric acid and other parameters rather than deriving from the
direct relationships between UVB/global radiation and OH radical. Under this circumstance, scaling factors $k_2$ and
$k_3$ are influenced by measured sulfuric acid, UVB and global radiation as well as calculated CS, which may
introduce substantial uncertainties. Moreover, for both concentration estimation and linearity (R, exponent and
linear slope), proxies with fitted scaling factors performed worse that those with constant scaling factors. This may
be due to the absence of linear relationships between proxies and photochemical terms (OH radical, UVB, or global
radiation), since the fitted scaling factors $k_1$, $k_2$ and $k_3$ all include the photochemical term (Table 3).





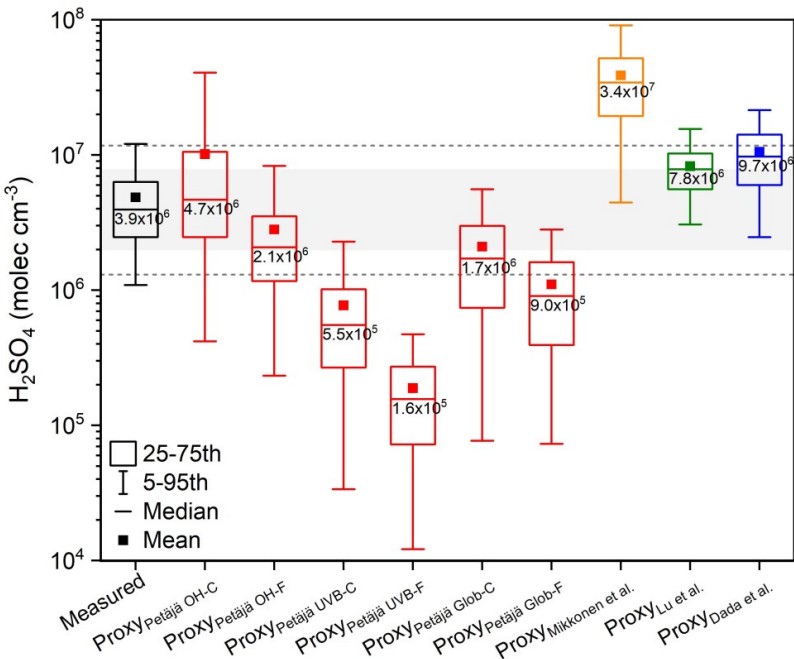

**Figure 3**. Sulfuric acid concentrations from measurement and estimated by proxies from literatures during the time window of 10:00-14:00 in 2019 (1st January to 31st December). Black value inside each box is the median concentration. "Proxy$_{Petäjä OH-C}$" "Proxy$_{Petäjä OH-F}$" "Proxy$_{Petäjä UVB-C}$" "Proxy$_{Petäjä UVB-F}$" "Proxy$_{Petäjä Glob-C}$" and "Proxy$_{Petäjä Glob-F}$" represent sulfuric acid proxies from the work of Petäjä et al., 2009, with $P_1$ proxy using OH radical with constant pre-factor $k_1$, $P_1$ proxy using OH radical with fitted pre-factor $k_1$, $P_2$ proxy using UVB with constant pre-factor $k_2$, $P_2$ proxy using UVB with fitted pre-factor $k_2$, $P_3$ proxy using global radiation with constant pre-factor $k_3$, and $P_3$ proxy using global radiation with fitted pre-factor $k_3$, respectively. "Proxy$_{Mikkonen et al.}$" "Proxy$_{Lu et al.}$" and "Proxy$_{Dada et al.}$" are sulfuric acid proxies from the work of Mikkonen et al. 2011, Lu et al. 2019 and Dada et al. 2020, respectively. Gray area covers 50% to 200% of median concentration of measured sulfuric acid, and two gray lines cover 33.3% to 300% of median concentration of measured sulfuric acid.

**Table 4.** The correlation coefficients (R), power exponents (Exponent) and slopes (Linear Slope) of the linear fittings between measured sulfuric acid concentration and the estimated ones using proxies from literatures, the relative errors (RE) of the estimated sulfuric acid concentrations to the measured one, as well as the ratios of proxy concentrations to measured concentration using mean ([Proxy/Measured]$_{mean}$) and median ([Proxy/Measured]$_{median}$) values. All parameters are fitted with Bisquare fitting.

| Proxy | R | Exponent | Linear Slope | RE (%) | [Proxy/Measured]$_{mean}$ | [Proxy/Measured]$_{median}$ | References |
|---|---|---|---|---|---|---|---|
| Proxy$_{Petäjä OH-C}$ | **0.97** | **1.17** | **0.85** | 74 % | **2.09** | **1.18** | |
| Proxy$_{Petäjä OH-F}$ | **0.78** | **0.80** | 0.30 | **35 %** | **0.58** | **0.53** | |
| Proxy$_{Petäjä UVB-C}$ | **0.84** | **1.08** | 0.09 | 57 % | 0.16 | 0.14 | (Petäjä et al., 2009) |
| Proxy$_{Petäjä UVB-F}$ | 0.52 | 0.63 | 0.01 | 64 % | 0.04 | 0.04 | |
| Proxy$_{Petäjä Glob-C}$ | 0.59 | **0.86** | 0.16 | **40 %** | 0.43 | 0.43 | |
| Proxy$_{Petäjä Glob-F}$ | 0.44 | 0.57 | 0.05 | 50 % | 0.23 | 0.23 | |
| Proxy$_{Mikkonen et al.}$ | 0.46 | 0.49 | 2.40 | 565 % | 7.98 | 8.70 | (Mikkonen et al., 2011) |
| Proxy$_{Lu et al.}$ | 0.38 | 0.30 | 0.42 | 91 % | **1.70** | **1.99** | (Lu et al., 2019) |
| Proxy$_{Dada et al.}$ | 0.18 | 0.30 | 0.45 | 133 % | 2.17 | 2.46 | (Dada et al., 2020) |

Note that the scaling factor $k_1$ of the OH-based proxy was obtained by replacing the left hand side of the equation with measured sulfuric acid concentration (Petäjä et al., 2009). Thus, $k_1$ was not derived from the chemical



production pathways of sulfuric acid, and the best-fit value of $k_1$ may vary from site to site. This limitation restricts
its applicability across a broader range of sites. Therefore, in this study, we aim to derive a proxy based entirely on
the formation and loss pathways of sulfuric acid, where the parameters, related pre-factors and exponents all have
chemical and physical meanings. Proxies of this kind should be applicable across different sites, since no site-
dependent scaling factors or exponents are used.
**3.3 Derivation of sulfuric acid proxies from its budget analysis**
During daytime, the main formation pathway of sulfuric acid is the $SO_2$ oxidation by OH radical, followed by
$O_2$ and $H_2O$ addition (R1–R3) (Finlayson-Pitts and Pitts Jr., 2000):
$$SO_2 + OH \xrightarrow{k_1} HSO_3 \qquad\qquad R1$$
$$HSO_3 + O_2 \xrightarrow{k_2} SO_3 + HO_2 \qquad\qquad R2$$
$$SO_3 + 2H_2O \xrightarrow{k_3} H_2SO_4 + H_2O \qquad\qquad R3$$
As OH radical oxidation is the rate-limiting step, the production rate of sulfuric acid is nearly equivalent to that of
$HSO_3$ and can be calculated as follows:
$$P_{[H_2SO_4]} = P_{[HSO_3]} = k_1 \cdot [SO_2] \cdot [OH]$$
Regarding sulfuric acid losses, the main loss pathway is its condensation sink onto particle surfaces (Dada et al.,
2020;Guo et al., 2021;Yang et al., 2021a), which can be written as:
$$L_{[H_2SO_4]} = [H_2SO_4] \cdot CS$$
The production and loss rates of sulfuric acid are much faster than its net concentration change (Guo et al., 2021),
so a pseudo-steady-state assumption can be applied:
$$k_1 \cdot [SO_2] \cdot [OH] \approx [H_2SO_4] \cdot CS$$
Then, the steady-state concentration of sulfuric acid can be estimated, which can be called as the OH-CS based
proxy:
$$Proxy_{OH,CS} = [H_2SO_4] = \frac{k_{SO_2-OH} \cdot [SO_2] \cdot [OH]}{CS} \qquad (1)$$
Here, $k_{SO_2-OH}$ is the rate constant of $SO_2$ oxidation by OH radical. It is taken as $1.3 \times 10^{-12}$ $(T/300)^{-0.7}$ cm$^3$ s$^{-1}$, where
T is the temperature in Kelvin (Wine et al., 1984;Atkinson et al., 2004), $[SO_2]$ and $[OH]$ are concentrations of $SO_2$
and OH radical in molec cm$^{-3}$, and CS is condensation sink of sulfuric acid in s$^{-1}$. Compared with the proxy proposed
by Petäjä et al. (2009), the pre-factor $k_{SO_2-OH}$ is not obtained by parameter fitting but is a verified reaction
coefficient derived from experiments. Therefore, this proxy is chemically meaningful and has the potential to be
used at various sites.
It is widely acknowledged that the OH radical is difficult to measure. Therefore, for most sites lacking OH
radical measurements, the OH-CS based proxy cannot be applied. A major production pathway for OH radical is
the photolysis of $NO_2$ and $O_3$, along with radical recycling (Lu et al., 2012;Ma et al., 2022), all driven by solar




radiation (Rohrer and Berresheim, 2006). Thus, UVB, a readily available parameter, can replace [OH] in equation
(1) to derive the second proxy as follows:
$$\text{Proxy}_{\text{UVB,CS}} = \frac{k_{\text{UVB−CS}} \cdot [\text{SO}_2] \cdot \text{UVB}}{\text{CS}} \quad (2)$$

where $k_{\text{UVB−CS}}$ is the pre-factor, and $[\text{SO}_2]$, UVB, and CS are in the units of molec cm$^{-3}$, W m$^{-2}$ and s$^{-1}$, respectively.
As shown in Figure S7A, OH radical and UVB has a linear correlation with R value of 0.86. The ratio of OH radical
to UVB is $6.14 \times 10^6$ molec cm$^{-3}$ W$^{-1}$ m$^2$. Accounting for this ratio yields $k_{\text{UVB−CS}}$ of $7.98 \times 10^{-6}$ (T/300)$^{-0.7}$ W$^{-1}$ m$^2$
s$^{-1}$. Replacing the left hand side of equation (2) with measured sulfuric acid concentration yields $k_{\text{UVB−CS}}$ of $7.5 \times$
$10^{-6}$ (T/300)$^{-0.7}$ W$^{-1}$ m$^2$ s$^{-1}$, which is close to the value derived from the OH-UVB relationship. This $k_{\text{UVB−CS}}$ is
finally used as it brings less deviation between measured and estimated sulfuric acid concentrations.
Furthermore, calculating CS requires particle size distribution data, which is not always available. In this case,
a surrogate parameter for CS is needed. The condensation sink of gaseous species onto particles is mainly
determined by the aerosol surface area. PM$_{2.5}$ measures the masses of particles. In principle, CS and PM$_{2.5}$ should
follow a power-law relationship with an exponent of 2/3. As expected, PM$_{2.5}^{2/3}$ and CS are well linearly correlated
(Figure S7B, R=0.92). Thus, replacing CS in equation (2) with PM$_{2.5}^{2/3}$ yields the third proxy as follows:
$$\text{Proxy}_{\text{UVB,PM}_{2.5}} = \frac{k_{\text{UVB−PM}_{2.5}} \cdot [\text{SO}_2] \cdot \text{UVB}}{\text{PM}_{2.5}^{2/3}} \quad (3)$$

where $k_{\text{UVB−PM}_{2.5}}$ is the pre-factor, and $[\text{SO}_2]$, UVB, and PM$_{2.5}$ are in the units of molec cm$^{-3}$, W m$^{-2}$ and μg m$^{-3}$,
respectively. The slope of CS to PM$_{2.5}^{2/3}$ is $2.67 \times 10^{-3}$ s$^{-1}$ μg$^{-2/3}$ m$^2$. Then, substituting [OH] with UVB and CS with
PM$_{2.5}^{2/3}$ yields $k_{\text{UVB−PM}_{2.5}}$ of $2.99 \times 10^{-3}$ μg$^{2/3}$ W$^{-1}$. Replacing the left hand side of equation (3) with measured
sulfuric acid concentration yields $k_{\text{UVB−PM}_{2.5}}$ of $2.8 \times 10^{-3}$ μg$^{2/3}$ W$^{-1}$, which is close to the value derived from the
OH-UVB and CS-PM$_{2.5}$ relationships and is finally used.
We summarize the three proxies incorporating the corresponding parameters as follows:
$$\text{Proxy}_{\text{OH,CS}} = (1.3 \times 10^{-12}) \times \left(\frac{T}{300}\right)^{-0.7} \times [\text{SO}_2] \times [\text{OH}] \div \text{CS} \quad (4)$$

$$\text{Proxy}_{\text{UVB,CS}} = (7.5 \times 10^{-6}) \times \left(\frac{T}{300}\right)^{-0.7} \times [\text{SO}_2] \times \text{UVB} \div \text{CS} \quad (5)$$

$$\text{Proxy}_{\text{UVB,PM}_{2.5}} = (2.8 \times 10^{-3}) \times [\text{SO}_2] \times \text{UVB} \div \text{PM}_{2.5}^{2/3} \quad (6)$$

The uncertainties of the OH-CS, UVB-CS, and UVB-PM$_{2.5}$ based proxies, based on equation (4)–(6), are estimated
to be 41.7%, 96.1%, and 100.4%, respectively. Details are provided in Section S3.
**3.4 Evaluation of different sulfuric acid proxies in this study**
**3.4.1 Performance of sulfuric acid proxies at Beijing Site**
Figure 4 shows the overall concentrations of measured and estimated sulfuric acid from proxies. The estimated
sulfuric acid concentrations from three proxies are generally in good agreement with the measured one, although
the OH-CS-based proxy yields slightly lower concentration than measurement. Additionally, the concentration




ranges estimated by proxies are broader than the measured one. Detailed sulfuric acid concentrations, including
mean, standard deviation, median, lower quartile and upper quartile values are summarized in Table S4.

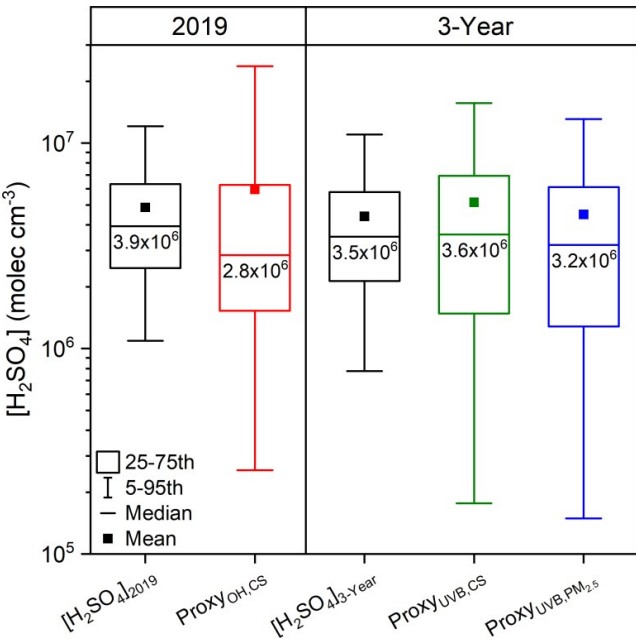

**Figure 4.** Sulfuric acid concentrations from measurement and estimated by proxies in this study during daytime (10:00-14:00).

380        The scatter plots of three proxies vs. measured sulfuric acid are shown in Figure 5. For all three proxies, the

estimated sulfuric acid concentrations are well correlated with the measured one, with most data points falling on
or near the 1:1 line. This suggests that the three steady-state based proxies generally perform well in estimating
daytime sulfuric acid concentration. However, slight deviations between the least-square-fit lines and the 1:1 lines
can be observed. To better understand these deviations, we summarize the correlation coefficients and power
exponents of the fits between measured and estimated sulfuric acid concentrations, as well as the relative errors of
the estimated concentrations (Table 5). The OH-CS-based proxy shows the best correlation ($R = 0.96$). The R values
for the UVB-CS based proxy (0.83) and the UVB-PM$_{2.5}$ based proxy (0.79) are also close to unity. The OH-CS
based proxy has an exponent of 1.14, indicating that the relationship between proxy and measured sulfuric acid is
not strictly linear, which could, to some extent, arise from the uncertainty in OH radical modelling. The exponents
of UVB-CS based proxy (1.02) and UVB-PM$_{2.5}$ based proxy (1.02) are very close to 1.0, suggesting excellent good
linear relationships between proxies measured sulfuric acid. The relative errors of three proxies are all within 50%,
which performs better than most proxies from previous studies (Table 4). Moreover, the ratios of proxy to measured
concentrations give the same result that they are in the range of 0.72–1.22, much closer to 1.0 than most proxies
from previous studies (Table 4).





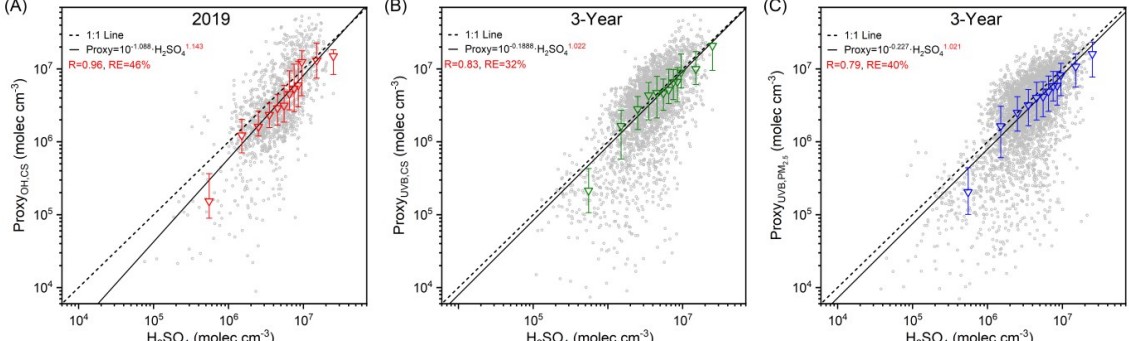

**Figure 5.** Sulfuric acid concentrations estimated by proxies in this study vs. the measured concentration during daytime (10:00-14:00) for (A) OH-CS based proxy in 2019, (B) UVB-CS based proxy in 3 years, and (C) UVB-PM$_{2.5}$ based proxy in 3 years. The black dashed lines are 1:1 lines, and the black lines are the distance weighted least square fits between proxy and measured sulfuric acid. Corresponding functions of the fits, correlation coefficients (R) and relative errors (RE) are shown in the legend. The triangle marker represents the binned data, where the up line, middle marker and bottom lines stand for upper quartile, median and lower quartile, respectively.

**Table 5.** The correlation coefficients (R) and power exponents (Exponent) of the linear fittings between measured sulfuric acid concentration and the estimated ones using proxies in this study, the relative errors (RE) of the estimated sulfuric acid concentrations to the measured one, as well as the ratios of proxy concentrations to measured concentration using mean ([Proxy/Measured]$_{mean}$) and median ([Proxy/Measured]$_{median}$) values.

| Year | Parameters | R | Exponent | RE (%) | [Proxy/Measured]$_{mean}$ | [Proxy/Measured]$_{median}$ |
|---|---|---|---|---|---|---|
| 2019 | Proxy$_{OH,CS}$ | 0.96 | 1.14 | 46% | 1.22 | 0.72 |
| 3-Year | Proxy$_{UVB,CS}$ | 0.83 | 1.02 | 32% | 1.17 | 1.03 |
| | Proxy$_{UVB,PM_{2.5}}$ | 0.79 | 1.02 | 40% | 1.02 | 0.91 |

To have better understanding on the performance of sulfuric acid proxies at any given moment, the time variations of sulfuric acid concentrations from three proxies and measurement are shown in Figures S8 and S9. Generally, the OH-CS based proxy provides a good estimation on daytime sulfuric acid concentration (Figure S8). Specifically, in 2019, the concentration estimated by this proxy matches well with the measured one in January, February, March, April, August, and September. In other months of 2019, it underestimates or overestimates sulfuric acid concentration. This shows that although the OH-CS-based proxy generally performs well, sulfuric acid concentration at a given moment may deviate. Similarly, sulfuric acid concentrations estimated by UVB-CS based and UVB-PM$_{2.5}$ based proxies generally match well with the measured one at most of the daytime, with deviations noticeable in several months over 3 years (Figures S8 and S9). These time variations are consistent with the findings in Section 3.1.2: the daily peak width of OH-CS based proxy is narrower than that of measured sulfuric acid, and the daily peak widths of UVB-CS based and UVB-PM$_{2.5}$ based proxies are narrower than that of OH-CS based proxy. Furthermore, the OH-CS based proxy partially reproduces the formation of sulfuric acid at night and early morning, with evidence on most days of January 2019 and some days of February 2019. Although UVB-CS based and UVB-PM$_{2.5}$ based proxies cannot estimate nighttime sulfuric acid, they provide a convenient, reliable and, more importantly, feasible way to trace the long-term daytime sulfuric acid concentration for sites without OH radicals.

Sulfuric acid concentration is estimated using OH radical, UVB, SO$_2$, CS and PM$_{2.5}$. We then use these parameters to assess how well the proxy-estimated concentrations match the measured values, and to determine the applicable parameter ranges of the proxies. Figure 6 shows that when [OH] is lower than $4 \times 10^5$ molec cm$^{-3}$, UVB





is lower than 0.10 W m$^{-2}$ or SO$_2$ is lower than 0.5 ppb, all three steady-state based proxies underestimate sulfuric
acid concentration. This suggests that when the OH radical, UVB, or SO$_2$ is low, other SO$_2$ oxidation pathways or
additional sulfuric acid sources contribute more to sulfuric acid formation. As [OH] increases, the ratio of proxy to
measured sulfuric acid gradually rises above 1.0. These deviations of OH-CS based proxy may arise from
uncertainties in OH radical modelling. As UVB and SO$_2$ increase, the ratios of proxies to measured sulfuric acid
stabilize around 1.0. This suggests that although the OH-CS based proxy is derived entirely from sulfuric acid
budget analysis, its long-term stability may not be as good as that of UVB-CS based or UVB-PM$_{2.5}$ based proxies,
given the intrinsic uncertainty in OH modeling. The ratio of UVB-CS based proxy stays around 1.0 when CS is
lower than 0.07 s$^{-1}$, accounting for ~96.3% of total data. Similarly, the ratio of UVB-PM$_{2.5}$ based proxy shows no
clear dependence on PM$_{2.5}$ when it is lower than 200 μg m$^{-3}$, accounting for ~99.6% of all datasets. This indicates
that these two proxies can be applied across almost all CS and PM$_{2.5}$ ranges. For OH-CS based proxy, sulfuric acid
concentration is underestimated when CS is lower than 0.015 s$^{-1}$ (~32.2%) or higher than 0.07 s$^{-1}$ (~3.6%). Higher
CS is also associated with more polluted conditions when other sulfuric acid sources such as primary emissions
may exist (Yang et al., 2021a). At lower CS, UVB-CS based proxy performs well, while OH-CS based proxy does
not, suggesting that slightly poor performance of OH-CS based proxy may arise from OH radical modelling.
Meanwhile, the performances of three steady-state based proxies show a clear dependence on RH. When RH is
lower than 60%, the ratios of proxies to measured sulfuric acid stabilize around 1.0. When RH exceeds 60% (~13.6%
of total data), these ratios increase with RH. Higher RH correlates with precipitation events with lower UVB and
lower SO$_2$, increasing the contribution of additional sulfuric acid sources. This may partly explain the
underestimation of proxies at higher RH.



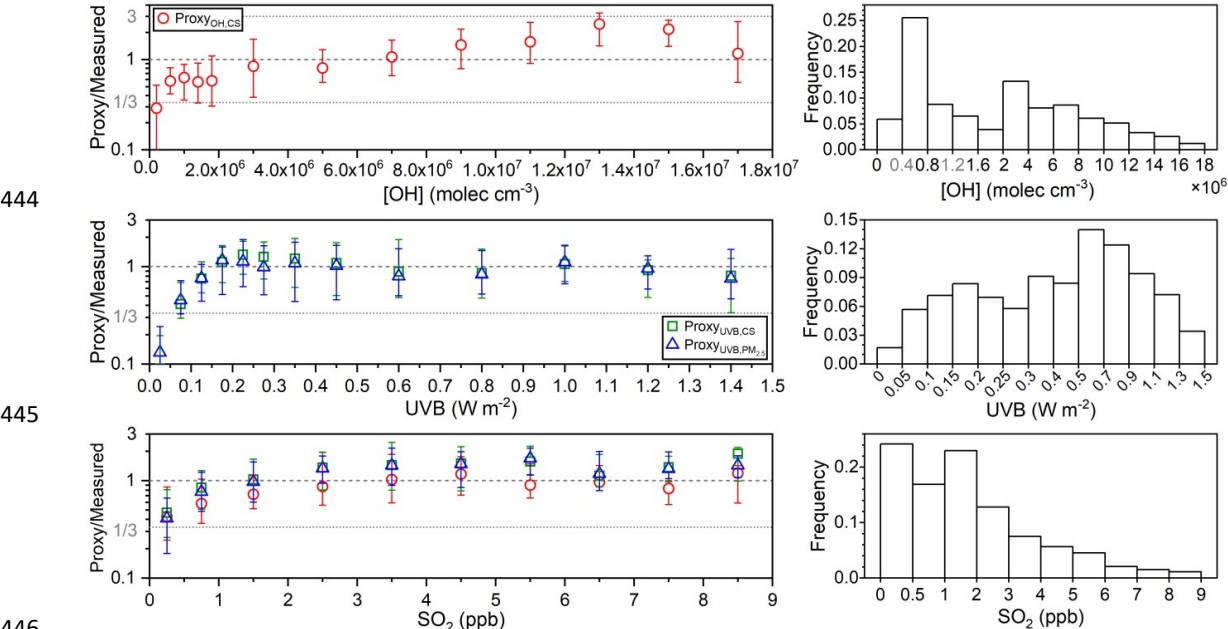



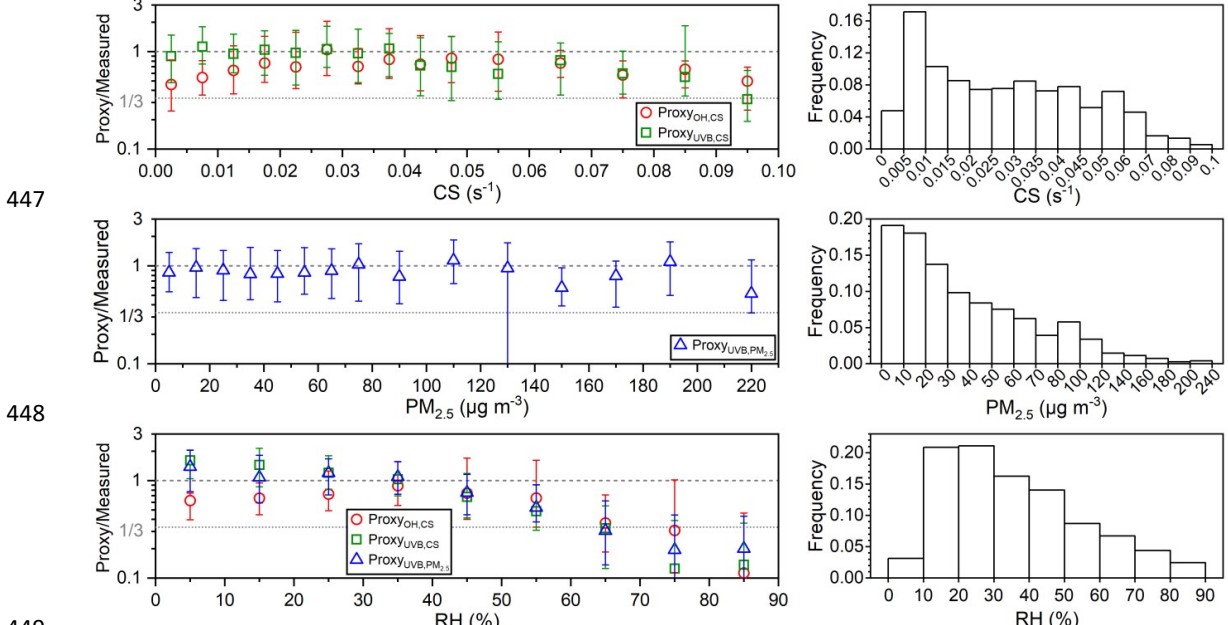

**Figure 6.** Left: The ratios of sulfuric acid concentrations estimated by proxies in this study to the measured one (Proxy/Measured) vs. concentration of OH radical ([OH]), UVB, $SO_2$, CS, $PM_{2.5}$ and RH during daytime (10:00-14:00) of 2019. Different colored markers represent different proxies. The up line, middle marker and bottom line stand for upper quartile, median and lower quartile values respectively. Right: Frequency distributions of corresponding parameters.

As shown in Figure S11, restricting the analysis to data within the optimal parameter ranges reduces the number of data points that deviate from the 1:1 line and have extremely low estimated sulfuric acid concentrations. Meanwhile, the correlation coefficients between the estimated and measured sulfuric acid concentrations generally improved, while the relative errors increased, and the improvement in the slopes of linear fits was not significant. This suggests that data outside the optimal parameter ranges generally have little impact on the fitting results.

### 3.4.2 Performance of sulfuric acid proxies at Hyytiälä, Finland

Because the three proxies above are derived from the budget analysis of sulfuric acid, equation (4)–(6) and their pre-factors should be applicable to other sites. To demonstrate this, we use datasets from a boreal forest site in Hyytiälä, Finland as test data. Figure 7 shows the scatter plots of UVB-CS based and UVB-$PM_{2.5}$ based proxies vs. measured sulfuric acid. For both proxies, most data points lie on or are near the 1:1 line, with R values close to 1.0, indicating good linear correlations between the estimated and measured sulfuric acid concentrations. The relative errors for UVB-CS based and UVB-$PM_{2.5}$ based proxies of Hyytiälä site are 97% and 80%, respectively, which are only slightly larger than those of the Beijing site (Table 5) but still within an acceptable range. The above results suggest that both proxies perform well in estimating daytime sulfuric acid concentration at Hyytiälä.

For the UVB-CS based proxy, the pre-factor $k_{UVB-CS}$ in equation (5) was chosen the same as Beijing. This proxy estimates sulfuric acid concentrations well at both Beijing and Hyytiälä sites using the same $k_{UVB-CS}$ value, indicating that the OH–UVB relationships, or the $k^{'}$ values in $[OH] = k^{'} \cdot UVB$, do not differ significantly between



these two sites. This further suggests that k' values at other sites should not differ significantly, and that $k_{UVB-CS}$
values should be similar across sites.

473       For the UVB-PM$_{2.5}$ based proxy, the pre-factors $k_{UVB-PM_{2.5}}$ in equation (6) are $2.8 \times 10^{-3}$ $\mu g^{2/3}$ W$^{-1}$ and $4.7 \times 10^{-3}$

$\mu g^{2/3}$ W$^{-1}$ for Beijing and Hyytiälä, respectively. This difference in $k_{UVB-PM_{2.5}}$ arises from the disparity of pre-
factor in CS = k · PM$_{2.5}^{2/3}$, where the values of k are $2.67 \times 10^{-3}$ $\mu g^{-2/3}$ m$^2$ and $1.59 \times 10^{-3}$ $\mu g^{-2/3}$ m$^2$ for Beijing (Figure
S7B) and Hyytiälä (Figure S12A), respectively. Specifically, the ratios of $4.7 \times 10^{-3}$ to $2.8 \times 10^{-3}$ and of $2.67 \times 10^{-3}$ to
$1.59 \times 10^{-3}$ are both 1.68. Therefore, considering the CS–PM$_{2.5}$ relationships, equation (6) is also applicable to
Hyytiälä. This tells us that when using the UVB-PM$_{2.5}$ based proxy to estimate sulfuric acid concentration, the k
value should be determined first to correct $k_{UVB-PM_{2.5}}$. Figures S8B–C show that the slope of CS to PM$_{2.5}^{2/3}$ is
stable across years and seasons at a given site. Therefore, by conducting short-term synchronous measurement of
PM$_{2.5}$ and particle size distribution, a reliable k can be obtained. In summary, these steady-state based proxies are
transferable proxies that can be widely used to estimate daytime sulfuric acid concentration at other atmospheric
sites.

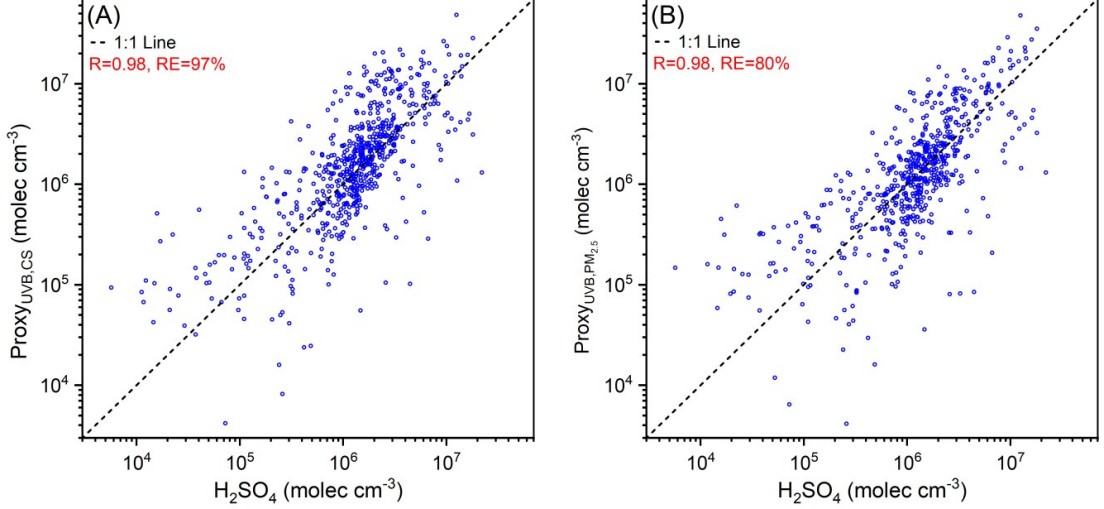

**Figure 7.** (A) UVB-CS based proxy (Proxy$_{UVB,CS}$) and (B) UVB-PM$_{2.5}$ based proxy (Proxy$_{UVB,PM_{2.5}}$) vs. measured sulfuric acid
of Hyytiälä, Finland during daytime (10:00-14:00) from 8$^{th}$ March to 13$^{th}$ Aug. 2018. The pre-factor of Proxy$_{UVB,CS}$ ($k_{UVB,CS}$)
is $7.5 \times 10^{-6}$ (T/300)$^{-0.7}$ W$^{-1}$ m$^2$ s$^{-1}$, which is the same as Beijing. The pre-factor of Proxy$_{UVB,PM_{2.5}}$ ($k_{UVB,PM_{2.5}}$) is $4.7 \times 10^{-3}$ $\mu g^{2/3}$
W$^{-1}$. In both two plots, the black dashed lines are 1:1 lines. Correlation coefficients (R) and the relative errors (RE) are shown
in the legend.

## 4. Summary and Conclusions

491       In this study, long-term measurement of sulfuric acid from 2019 to 2021 was conducted in urban Beijing.

Daytime sulfuric acid concentration ranges from $2.0 \times 10^6$ to $7.4 \times 10^6$ molec cm$^{-3}$ and shows a general declining trend,
with an average annual decrease of 14%, which is mainly due to SO$_2$ reduction. In addition, sulfuric acid
concentration shows a clear seasonal variation that tracks UVB, reaching the highest in May and September and
decreasing to the lowest from November to February of next year. In July and August, frequent precipitation lowers



UVB and $SO_2$, resulting in lower sulfuric acid. Nighttime sulfuric acid concentration ranges from $1.6\times10^5$ to $6.3\times10^5$
molec $cm^{-3}$, about one order of magnitude lower than daytime. In warmer seasons, the sources of nighttime sulfuric
acid, such as benzene-related emissions and alkene ozonolysis, are stronger, and the losses are weaker, leading to
higher sulfuric acid level. The diurnal variations of photo-oxidation related parameters deviate slightly from sulfuric
acid. Sulfuric acid peaks earliest, followed by $J(NO_2)$, $J(O^1D)$, UVB, global radiation, and OH radical. Meanwhile,
the peak width of sulfuric acid is the widest, followed by $J(NO_2)$, global radiation, OH radical, $J(O^1D)$, and UVB.
The challenges in sulfuric acid measurement hinder its widespread observation. To obtain sulfuric acid proxies
applicable to most sites, we derive three sulfuric acid proxies directly from its steady-state budget analysis, named
as OH-CS based, UVB-CS based, and $UVB-PM_{2.5}$ based proxies. All three proxies perform well in estimating
sulfuric acid concentration during 10:00–14:00. We also evaluate the performance of nine sulfuric acid proxies
proposed in previous studies: seven based on formation and loss pathways (Petäjä et al., 2009;Dada et al., 2020)
and two derived from numerical regression (Mikkonen et al., 2011;Lu et al., 2019). Results show that $Proxy_{Petäjä\ OH-}$
$_C$ and $Proxy_{Petäjä\ OH-F}$ generally reproduce daytime sulfuric acid concentrations well, with estimated concentrations
closet to the measured one, correlation coefficients being 0.97 and 0.78, respectively, and relative errors being 74%
and 35%, respectively. However, the scaling factors therein are obtained by fitting the proxy equations. Thus, these
scaling factors are influenced by measurement reliability and have limited applicability at other sites. By contrast,
our proxies are derived directly from sulfuric acid budget analysis, and the parameters in the proxy equations are
transferable that can be used at a boreal forest site in Hyytiälä, Finland. Therefore, the three proxies developed in
this study have high potential for estimating daytime sulfuric acid concentrations at various sites.
It should be noted that the OH radical used in this study is not measured, but derived from a model simulation.
Under this circumstance, the OH-CS based proxy generally performs well, but has some deviations when OH radical
is in the range of $1.2–1.6\times10^7$ molec $cm^{-3}$ and CS is lower than $0.015\ s^{-1}$. Although three steady-state-based proxies
generally perform well, they are not suitable under certain conditions. When OH radical, UVB and $SO_2$ are too low,
when CS and $PM_{2.5}$ are too high, or when RH exceeds 60%, estimated sulfuric acid concentration may deviate from
the actual concentration to a larger extent. Moreover, three proxies cannot fully reproduce sulfuric acid
concentration in early morning and at nightfall. This indicates that during these two periods, other sulfuric acid
sources, such as direct emission, alkenes ozonolysis and other formation pathways, are also important.
Here are some suggestions for the selection of three proxies. If one site has comprehensive measurement of OH
radical, particle size distribution and $SO_2$, the OH-CS based proxy illustrated by equation (4) is preferred, since it
estimates daytime concentration well and partly captures diurnal variation and nighttime sulfuric acid. Moreover,
the pre-factor in equation (4) is the actual $OH + SO_2$ reaction rate, making it suitable to all atmospheric sites. Then,
if OH radical is not directly measured, but UVB, $SO_2$, and particle size distribution are available, the UVB-CS
based proxy illustrated by equation (5) is preferred. Although it cannot perfectly trace the diurnal variation of
sulfuric acid, it estimates daytime concentration well. Moreover, because its pre-factor is transferable, it is
convenient and straightforward to use. Finally, if neither OH radical nor particle size distribution is measured, but



UVB, $SO_2$, and $PM_{2.5}$ are available, the UVB-$PM_{2.5}$ based proxy should be the right choice. These three parameters
used are commonly measured, giving this proxy broad applicability. Noted that $k_{UVB-PM_{2.5}}$ in equation (6) various
across sites. For better accuracy, short-term synchronous measurement of particle size distribution and $PM_{2.5}$ is
suggested for obtaining the pre-factor (k) in $CS = k \cdot PM_{2.5}^{2/3}$ and then correcting $k_{UVB-PM_{2.5}}$.
The acquisition of fundamental sulfuric acid concentration datasets is of great significance for elucidating the
global spatial distribution and long-term temporal trends of sulfuric acid. This may further promote researches on
the mechanisms of atmospheric nucleation, cluster growth, secondary aerosol formation, and pollution event
evolution at corresponding regions.

**Data and materials availability:** Datasets for this paper can be accessed at https://zenodo.org/records/17216660
(Guo et al., 2025).

**Author contributions:**
YG, CY and YL designed the study and wrote the paper. CL, CD, YZhang, YZhou, XC, WM, NS, ZL, CH, XF,
FZ, ZF, ZW, and YZ conducted the measurement and collected the data. HZ and YJ did the modelling. JJ, BZ and
MK are acknowledged for valuable suggestions. And co-authors have read and commented on the paper.

**Competing interests:** The authors declare that they have no conflict of interests.

**Acknowledgements and funds:** This study is funded by the National Natural Science Foundation of China (NSFC)
(grant No. 22327806), the Science and Technology Project of Hebei Education Department (grant No. QN2025049),
and the Doctoral Fund of Hebei Vocational University of Industry and Technology (No. bz202402).



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
