# Peer review of "Measurement report: Three-year characteristics of sulfuric acid in"

_EGUsphere, 2025_

## Author Comment (AC1)

**Response to Reviewers**

**Referee #1**

Guo et al. present a comprehensive analysis of a three-year sulfuric acid concentration dataset at an urban site and put forward three daytime concentration proxies. Compared to proxies presented in earlier research, the three proxies in this study exhibit better results in reproducing the daytime sulfuric acid concentration. Furthermore, the authors discuss the possibility of applying the three proxies in this study to other environments.

In the atmospheric chemistry research area, especially for atmospheric new particle formation research, sulfuric acid is a critical substance as it is the major oxidation product from sulfur dioxide and possesses non-volatile characteristics. However, measurement of sulfuric acid based on mass spectrometer instruments is not yet wide and continuous enough for us to generate a big picture of new particle formation in different environments and periods. The study from Guo et al. not only makes a valuable complement to the current sulfuric acid concentration dataset but also gives an attempt on sulfuric acid concentration estimation based on the work of predecessors. I think once the questions listed below are addressed, this paper is adequate to be published.

We thank the reviewer for the constructive comments and suggestions. The point-to-point response to the comments is given below. The comments, our replies, and corresponding changes in the revised manuscript and supplementary information are marked in black, blue, and green texts, respectively.

**--1.** I do not agree with the "applicable to various sites" description in the title. The three proxies in this study only consider the source of sulfur dioxide oxidation by OH radical and the authors only verify these proxies against the dataset in Hyytiälä. As mentioned in the introduction, for example, sulfuric acid may arise from dimethyl disulfide or dimethyl sulfide oxidation in coastal areas, and this has caused pronounced underestimation using a similar proxy (k [SO$_2$] [OH]/CS).

Response: Thank you for your valuable suggestion.

This study indeed only verifies the applicability of sulfuric acid proxies in urban Beijing and forest Hyytiälä. Therefore, the expression of "applicable to various sites" is inappropriate. After consideration, the title has been revised to "applicable to inland sites" to indicate that the sulfuric acid proxies proposed in this study can be used in non-coastal sites other than urban Beijing. (Line 3 in the main text, Page 1; and Line 4 in the supplement, Page 1)

**--2.** According to Fig 1C, Fig S2, Fig 6 and Fig S10, the current sulfur dioxide concentration is relatively low in Beijing (e.g., the median concentration from May to Nov. is all close to or lower than 0.5 ppb in 2021 even without precipitation data (Fig 1C and Fig S2), which can also be seen from the frequency plot of sulfur dioxide concentration in Fig 6 and Fig S10). While the three proxies in this study exhibit negative bias when sulfur dioxide concentration lies in this range (Fig 6 and Fig S10). As far as I know, new particle formation events (NPF) often correspond to scavenging conditions in polluted urban environments, which means that NPF are most likely to stay in the low CS and low sulfur dioxide concentration range. Despite the aforementioned negative bias in the low sulfur dioxide concentration range, the three proxies in this study still behave well for the lowest CS bin which corresponds to scavenging conditions. So, this present method confused me when it comes to how well these proxies work exactly during NPF since the authors highlight the importance of these proxies for NPF analysis.

If you could break the individual bins in the frequency plot into accumulated columns classified by NPF and non-NPF, I think the plot will be more straightforward and valuable for others. Or you can find other ways to sort out the inherent co-occurrence relationship between these parameters (UVB, SO$_2$ concentration, CS, RH).

Response: Thanks a lot for your suggestion.

As suggested, we have broken the individual bins in the revised frequency plots (Figures 6 and S10) into accumulated columns classified by "NPF", "Non-NPF", "Undefined", and "No Data" periods.

The performance of sulfuric acid proxies in estimating sulfuric acid concentration during NPF and Non-NPF periods still depends on the ranges of the parameters in proxy equations. On NPF days, the proxies underestimate sulfuric acid concentration when $SO_2$ is lower than 0.5 ppb; when $SO_2$ exceeds 0.5 ppb, the estimated concentration is close to the measured one. As shown in the revised Figures 6 and S10, about 30% of NPF cases fall outside the optimal range of $SO_2$, while most NPF cases fall within the optimal ranges of OH radical, UVB, CS, $PM_{2.5}$, and RH. Consequently, during NPF periods, the performance of three proxies mainly depends on the $SO_2$ concentration at that time.

[Figure]

[Figure]

**Figure 6.** Left: The ratios of sulfuric acid concentrations estimated by proxies in this study to the measured one (Proxy/Measured) vs. concentration of OH radical ([OH]), UVB, $SO_2$, CS, $PM_{2.5}$ and RH during daytime (10:00-14:00) of 2019. Different colored markers represent different proxies. The up line, middle marker and bottom line stand for upper quartile, median and lower quartile values respectively. Right: Frequency distributions of corresponding parameters classified by "NPF", "Non-NPF", "Undefined", and "No Data" periods.

**Figure S10.** Left: The ratios of sulfuric acid concentrations estimated by proxies in this study to the measured one (Proxy/Measured) vs. UVB, $SO_2$, CS, $PM_{2.5}$ and RH during daytime (10:00-14:00) of three years. Different colored markers

represent different proxies. The up line, middle marker and bottom line stand for upper quartile, median and lower quartile values respectively. Right: Frequency distributions of corresponding parameters classified by "NPF", "Non-NPF", "Undefined", and "No Data" periods.

To clarify this consideration, we also added the above discussions in the revised manuscript:

"Sulfuric acid is a key precursor in NPF processes. Therefore, it is necessary to assess how well these proxies perform during NPF periods. As shown in Figures 6 and S10, about 30% of NPF cases fall outside the optimal range of $SO_2$, while most NPF cases fall within the optimal ranges of OH radical, UVB, CS, $PM_{2.5}$, and RH. Consequently, during NPF periods, the performance of three proxies mainly depends on the $SO_2$ concentration at that time." (Line 465–468, Page 18)

**--3.** In Fig 5, as the dataset becomes larger, more points fall into the measured sulfuric acid concentration >> proxy calculated sulfuric acid concentration regime. What is the parameter condition of these deviated points? Does the parameter condition of these deviated points match the results you found "When OH radical, UVB and $SO_2$ are too low, when CS and $PM_{2.5}$ are too high, or when RH exceeds 60%, estimated sulfuric acid concentration may deviate from the actual concentration to a larger extent"? And why are there fewer points deviating into the measured sulfuric acid concentration << proxy calculated sulfuric acid concentration regime?

Response: As shown in Figure R1, cases with measured sulfuric acid concentration much larger than the proxy estimated concentrations occur mainly at RH larger than 60%. This result is consistent with our findings that "When OH radical, UVB and $SO_2$ are too low, when CS and $PM_{2.5}$ are too high, or when RH exceeds 60%, estimated sulfuric acid concentration may deviate from the actual concentration to a larger extent".

The proxies proposed in this study considers sulfuric acid generation via the OH-initiated oxidation of $SO_2$. Theoretically, if other sources contribute substantially to sulfuric acid formation, the proxies will underestimate its concentration. If the proxies overestimate sulfuric acid concentration considerably, it would imply that the production the $SO_2$+OH pathway far exceeds observation, which is actually unreasonable. Such overestimation is more likely due to measurement errors, such as the overestimation of OH radical, UVB or $SO_2$, or the underestimation of CS. Therefore, it is expected that cases with measured sulfuric acid concentration much lower than the estimated concentrations are rare. This also suggests that the measurement and calculation of those parameters are reliable.

[Figure]

**Figure R1.** Sulfuric acid concentrations estimated by proxies in this study vs. measured concentration during daytime (10:00-14:00) for (A) OH-CS based proxy in 2019, (B) UVB-CS based proxy in 3 years, and (C) UVB-$PM_{2.5}$ based proxy in 3 years. The black dashed lines are 1:1 lines, and the black lines are the distance weighted least square fits between proxy and measured sulfuric acid. Corresponding functions of the fits, correlation coefficients (R) and relative errors (RE) are shown in the legend. The triangle marker represents the binned data, where the up line, middle marker and bottom lines stand for upper quartile, median and lower quartile, respectively. The black square correspond to when RH is larger than 60%.

**--4.** In equation (6), why is there no term $(T/300)^{-0.7}$ anymore?

Response: In the revised manuscript, Eqns. (4)–(6) have been numbered as Eqns. (16)–(18), respectively.

$$\text{Proxy}_{\text{OH,CS}} = (1.3 \times 10^{-12}) \times \left(\frac{T}{300}\right)^{-0.7} \times [\text{SO}_2] \times [\text{OH}] \div \text{CS} \qquad (16)$$

$$\text{Proxy}_{\text{UVB,CS}} = (7.5 \times 10^{-6}) \times \left(\frac{T}{300}\right)^{-0.7} \times [\text{SO}_2] \times \text{UVB} \div \text{CS} \qquad (17)$$

$$\text{Proxy}_{\text{UVB,PM}_{2.5}} = (2.8 \times 10^{-3}) \times [\text{SO}_2] \times \text{UVB} \div \text{PM}_{2.5}{}^{2/3} \qquad (18)$$

For urban Beijing, the temperature term $(T/300)^{-0.7}$ is very close to 1 (Table R1). Thus, the temperature term has a negligible influence on the estimation of sulfuric acid concentration. Therefore, this term is removed from Eq. (18) to simply the UVB-PM$_{2.5}$ based proxy, which potentially has the widest applicability because it uses the routinely measured parameters. Under extreme temperatures of -30 and 50°C, this temperature term evaluates to 1.158 and 0.949, respectively. Compared with the uncertainty of this UVB-PM$_{2.5}$ based proxy using Eq. (18) (Section S3, 100.4%), the error introduced by removing the temperature term is acceptable.

**Table R1.** Statistical parameters of the temperature term $(T/300)^{-0.7}$ in Eqns. (16) and (17).

| Parameters | 2019 | 3-Year |
|---|---|---|
| Mean | 1.024 | 1.024 |
| Standard Deviation | 0.029 | 0.027 |
| Median | 1.022 | 1.021 |
| 25th Percentile | 0.998 | 1.000 |
| 75th Percentile | 1.049 | 1.047 |
| 5th Percentile | 0.986 | 0.988 |
| 95th Percentile | 1.071 | 1.070 |

**--5.** Some details in writing. For example, in Table 3, there is no parameter "f" in the Proxy$_{\text{Lu et al.}}$. Please check again.

Response: Thanks a lot for pointing out that.

The letter "f" should be the exponent of NO$_x$, and there is no parameter "e" in the Proxy$_{\text{Lu et al.}}$. We have corrected this error in Table 3, and double checked the entire revised manuscript.

**Table 3.** The equations and internal parameters of nine sulfuric acid proxies from literatures.

| Proxy | Equation | Parameters | Reference |
|---|---|---|---|
| Proxy$_{\text{Lu et al.}}$ | $k_0 \cdot \text{UVB}^a \cdot [\text{SO}_2]^b \cdot \text{CS}^c \cdot (\text{O}_3{}^d + \text{NO}_x{}^f)$ | $k_0$=0.0013, a=0.13, b=0.40, c=-0.17, d=0.44, f=0.41 | (Lu et al., 2019) |

**Referee #2**

This study presents long-term measurements of sulfuric acid ($H_2SO_4$) in urban Beijing (2019–2021) and develops three formation- and loss-based proxies (OH–CS, UVB–CS, and UVB–$PM_{2.5}$). All three proxies reproduced observed concentrations well, and the UVB–$PM_{2.5}$ proxy was successfully applied to a boreal forest site, indicating its potential for broader global application in studying atmospheric nucleation and aerosol growth. This paper presents innovative and valuable findings; however, some revisions are needed to improve clarity before publication.

We thank the reviewer for the constructive comments and suggestions. We have carefully revised our manuscript and supplement accordingly. The point-to-point response to the comments is given below. And the comments, our replies, and the corresponding changes in the manuscript and supplementary information are in black, blue, and green, respectively.

**Major Comments**

1. The manuscript would benefit from improved organization. It currently appears to have been written by two different contributors (modeling and observation parts), resulting in inconsistent flow and style. The overall readability should be improved by polishing the language and avoiding vague expressions.

Response: Thanks for your valuable suggestion.

The expression of the modelling part has been revised as follows:

"The Weather Research and Forecasting Model-Community Multiscale Air Quality (WRF-CMAQ) model was applied to simulate the concentration of OH radical, the photolysis rate of $NO_2$ ($J(NO_2)$) and the photolysis rate for producing excited atomic oxygen from $O_3$ ($J(O^1D)$). Simulations covered the period from 1st January, 2019 to 19th February, 2020. The physical options in WRF (version 3.9.1) were the same as in Zheng et al. (2019a). The CMAQ model (version 5.3.2) was coupled with the two-dimensional Volatility Basis Set (2D-VBS) (Zhao et al., 2016), where the SAPRC07 mechanism was adopted for gas-phase chemistry, and the AERO6 (Sarwar et al., 2011) was used for aerosol module. The modelling domain was the same as in Zheng et al. (2020), where the horizontal resolution was 27 km × 27 km and the vertical grid had 14 layers. Default planetary boundary layer settings were used. To minimize the influence of initial conditions, simulations were spun up 5 days before the modelling period." (Line 173–181, Page 6–7)

2. The modeled concentrations of key pollutants should be evaluated against available observations (e.g., $O_3$, $SO_2$, and other relevant species) to validate model performance and strengthen the credibility of the results.

Response: Thanks for the valuable comment.

The emission inventory and the WRF-CMAQ modeling system used in this study have been widely applied and extensively validated in previous studies using multiple lines of evidence, including ground-based monitoring networks and satellite retrievals (Zhao et al., 2018;Zheng et al., 2019a, b, 2023, 2024;Chang et al., 2023). For instance, the modeling system demonstrated good performance in reproducing the concentrations of several organic aerosol components during the COVID-19 period (Chang et al., 2023). These evaluations have consistently demonstrated that the modeling system is capable of reasonably reproducing the spatial and temporal variations of major air pollutants across China.

To further demonstrate the reliability of the results in the present study, we have added dedicated model-observation comparisons. Specifically, we included validation at the BUCT supersite corresponding to our study period, as well as nationwide evaluations against national monitoring stations for the full years of 2019 and 2020

(Tables S5–S7). The results show that simulated concentrations of key pollutants agree well with observations in both magnitude and temporal variability.

**Table S5.** Performance statistics for the comparison between simulated (SIM) and observed (OBS) concentrations of $NO_2$, $SO_2$, $O_3$, and $PM_{2.5}$ at BUCT site during the study period (2019.1.1 – 2020.3.15).

| Variables | $NO_2$ (ppb) | $SO_2$ (ppb) | $PM_{2.5}$ ($\mu g/m^3$) | $O_3$ (ppb) |
|---|---|---|---|---|
| Mean OBS | 19.5 | 1.6 | 27.7 | 45.2 |
| Mean SIM | 17.6 | 1.9 | 28.6 | 40.5 |
| Normalized Mean Bias | -10% | 33% | 3% | -11% |
| Normalized Mean Error | 63% | 88% | 66% | 57% |

**Table S6.** Performance statistics for the comparison between simulated (SIM) and observed (OBS) concentrations of $NO_2$, $SO_2$, $O_3$, and $PM_{2.5}$ at national monitoring sites in China in 2019.

| Variables | $NO_2$ (ppb) | $SO_2$ (ppb) | $PM_{2.5}$ ($\mu g/m^3$) | $O_3$ (ppb) |
|---|---|---|---|---|
| Mean OBS ($\mu g/m^3$) | 29.3 | 11.3 | 39.2 | 93.3 |
| Mean SIM ($\mu g/m^3$) | 26.2 | 8.5 | 34.3 | 92.9 |
| Normalized Mean Bias | -11% | -25% | -12% | 0% |
| Normalized Mean Error | 20% | 27% | 20% | 16% |

**Table S7.** Performance statistics for the comparison between simulated (SIM) and observed (OBS) concentrations of $NO_2$, $SO_2$, $O_3$, and $PM_{2.5}$ at national monitoring sites in China in 2020.

| Variables | $NO_2$ (ppb) | $SO_2$ (ppb) | $PM_{2.5}$ ($\mu g/m^3$) | $O_3$ (ppb) |
|---|---|---|---|---|
| Mean OBS ($\mu g/m^3$) | 26.0 | 10.0 | 34.5 | 91.6 |
| Mean SIM ($\mu g/m^3$) | 21.7 | 6.9 | 31.0 | 83.9 |
| Normalized Mean Bias | -17% | -31% | -10% | -8% |
| Normalized Mean Error | 28% | 38% | 22% | 19% |

We have added Table S5–S7 in the supplement and revised the manuscript as follows:

"This WRF-CMAQ model and the emission inventory have been widely applied and validated in previous studies using multiple lines of evidence, including ground-based monitoring networks and satellite retrievals (Zhao et al., 2018;Zheng et al., 2019a, b, 2023, 2024;Chang et al., 2023). Simulated concentrations of key pollutants agree well with observations in both magnitude and temporal variability (Tables S5–S7). These demonstrate that the modeling system reasonably reproduces the spatial and temporal variations of major air pollutants across China." (Line 182–186, Page 7)

**Minor Comments**

1. Please avoid vague wording such as "scarce".

Response: Thanks a lot for your suggestion.

We have revised the sentence containing the word "scarce" as follows:

"However, long-term measurement of it is only available at a few sites." (Line 22, Page 1)

2. Abbreviations (e.g., OH–CS, UVB–CS, UVB–$PM_{2.5}$) and variables (e.g., J($NO_2$)) in the abstract should be spelled out at first mention for clarity.

Response: Thanks a lot for pointing out that.

The journal *Atmospheric Chemistry and Physics* requires an abstract of fewer than 250 words. Currently, the abstract contains 247 words. We attempted to spell out all abbreviations and the parameter J($NO_2$). However, doing

so would exceed the word limit. After consideration, the abbreviations "(OH-CS, UVB-CS and UVB-PM$_{2.5}$ based proxies)" were removed, which does not change the original meaning. (Line 26, Page 1)

The full name of the parameter J(NO$_2$) has been added to the revised manuscript:

"… photolysis rate of NO$_2$ (J(NO$_2$)) …" (Line 31, Page 1)

In addition, the abbreviation for "condensation sink" in the abstract has been added to the revised manuscript:

"… condensation sink (CS) …" (Line 29, Page 1)

3. Lines 101–109: Please present each equation on a separate line and number them sequentially.

Response: Thanks a lot for your suggestion.

We have placed each equation on a separate line and numbered them sequentially in the revised manuscript.

$$L1 = B \cdot k \cdot \text{Radiation} \cdot [SO_2] \cdot CS^{-1} \tag{1}$$

$$L3 = B \cdot k \cdot \text{Radiation} \cdot [SO_2]^{0.5} \tag{2}$$

$$[H_2SO_4] = 0.0013 \cdot UVB^{0.13} \cdot [SO_2]^{0.40} \cdot CS^{-0.17} \cdot ([O_3]^{0.44} + [NO_x]^{0.41} \tag{3}$$

$$([H_2SO_4] = -\frac{CS}{2k_3} + \sqrt{\left(\frac{CS}{2k_3}\right)^2 + \frac{[SO_2]}{k_3}(k_1 \cdot \text{GlobRad} + k_2 \cdot [O_3][\text{Alkenes}]))} \tag{4}$$

$$[H_2SO_4] = \frac{k_1 \cdot \text{GlobRad}[SO_2] + k_2 \cdot [SO_2][O_3][\text{Alkene}]}{CS} \tag{5}$$

$$[H_2SO_4] = -\frac{CS}{2k_3} + \sqrt{\left(\frac{CS}{2k_3}\right)^2 + \frac{[SO_2]}{k_3}k_1 \cdot \text{GlobRad}} \tag{6}$$

4. Line 139: Ensure that all equations in the manuscript are numbered consistently and in order.

Response: Thank a lot.

We have double checked all equation numbers and revised some of them to ensure they are consistent and in order.

$$H_2SO_4 = \frac{HSO_4^- + H_2SO_4NO_3^-}{NO_3^- + HNO_3NO_3^- + (HNO_3)_2NO_3^-} \times C \tag{7}$$

$$CS = 4\pi D \int_0^{d_p max} \beta_m(d_p') d_p' N_{d_p'} dd_p' = 4\pi D \sum_{d_p'} \beta_{m,d_p'} d_p' N_{d_p'} \tag{8}$$

$$RE = \frac{1}{n} \cdot \sum_{i=1}^{n} \frac{|[H_2SO_4]_{pro,i} - [H_2SO_4]_{mea,i}|}{[H_2SO_4]_{mea,i}} \tag{9}$$

$$P_{[H_2SO_4]} = P_{[HSO_3]} = k_1 \cdot [SO_2] \cdot [OH] \tag{10}$$

$$L_{[H_2SO_4]} = [H_2SO_4] \cdot CS \tag{11}$$

$$k_1 \cdot [SO_2] \cdot [OH] \approx [H_2SO_4] \cdot CS \tag{12}$$

$$\text{Proxy}_{OH,CS} = [H_2SO_4] = \frac{k_{SO_2-OH} \cdot [SO_2] \cdot [OH]}{CS} \tag{13}$$

$$\text{Proxy}_{UVB,CS} = \frac{k_{UVB-CS} \cdot [SO_2] \cdot UVB}{CS} \tag{14}$$

$$\text{Proxy}_{UVB,PM_{2.5}} = \frac{k_{UVB-PM_{2.5}} \cdot [SO_2] \cdot UVB}{PM_{2.5}^{2/3}} \tag{15}$$

$$\text{Proxy}_{OH,CS} = (1.3 \times 10^{-12}) \times \left(\frac{T}{300}\right)^{-0.7} \times [SO_2] \times [OH] \div CS \tag{16}$$

$$\text{Proxy}_{\text{UVB,CS}} = (7.5 \times 10^{-6}) \times \left(\frac{T}{300}\right)^{-0.7} \times [SO_2] \times UVB \div CS \qquad (17)$$

$$\text{Proxy}_{\text{UVB,PM}_{2.5}} = (2.8 \times 10^{-3}) \times [SO_2] \times UVB \div PM_{2.5}{}^{2/3} \qquad (18)$$

5. Line174: Revise the sentence to remove the repeated name in parentheses — it should read: "The modelling domain was the same as in Zheng et al. (2020)."

Response: Thanks a lot for your valuable suggestion.

We have revised this sentence and others with the same issue.

"The physical options in WRF (version 3.9.1) were the same as in Zheng et al. (2019a)." (Line 176, Page 6)

"The modelling domain was the same as in Zheng et al. (2020), …" (Line 179, Page 7)

6. Line 197: In the sentence "Thus, the yearly decline of sulfuric acid is mainly attributed to the decrease of $SO_2$ (by ~25% per year)," please provide appropriate references to support this statement.

Response: Thanks a lot for your suggestion.

This yearly decline of $SO_2$ was derived from the $SO_2$ measurement in this study. To eliminate the seasonal influence, we first calculated the percentage decrease of $SO_2$ of each month in 2021 relative to 2019. We then averaged the 12 monthly percentages to obtain the annual $SO_2$ decline (Table S8). The annual percentage decline of sulfuric acid was calculated using the same procedure.

**Table S8**. Monthly concentration of $SO_2$ (ppb) during daytime (10:00-14:00) from 2019 to 2021. "NaN" means there is no data available.

| Month | 2019 | | | 2020 | | | 2021 | | | Annual decline using median value / % |
|---|---|---|---|---|---|---|---|---|---|---|
| | Median | 25th | 75th | Median | 25th | 75th | Median | 25th | 75th | |
| January | 3.00 | 1.51 | 5.71 | 3.63 | 1.94 | 5.15 | 1.50 | 0.94 | 2.66 | -24.9 |
| February | 2.07 | 0.95 | 4.41 | 1.75 | 0.51 | 3.20 | 2.03 | 1.17 | 3.00 | -1.0 |
| March | 3.61 | 0.78 | 5.40 | 1.21 | 0.45 | 2.67 | 1.07 | 0.63 | 1.90 | -35.2 |
| April | 2.08 | 0.79 | 3.54 | 1.88 | 0.47 | 3.94 | 1.02 | 0.36 | 2.08 | -25.4 |
| May | 2.18 | 0.55 | 3.22 | 2.36 | 0.92 | 3.15 | 0.51 | 0.24 | 1.07 | -38.3 |
| June | 1.88 | 0.70 | 3.10 | 0.58 | 0.29 | 1.91 | 0.30 | 0.16 | 0.60 | -41.9 |
| July | 0.25 | 0.16 | 0.79 | 0.22 | 0.07 | 0.48 | 0.08 | 0.06 | 0.10 | -34.7 |
| August | 0.36 | 0.28 | 0.87 | NaN | NaN | NaN | 0.23 | 0.18 | 0.32 | -18.9 |
| September | 1.05 | 0.79 | 1.33 | 0.40 | 0.22 | 0.73 | 0.27 | 0.11 | 0.40 | -37.0 |
| October | 0.77 | 0.31 | 1.70 | 0.68 | 0.46 | 1.49 | 0.35 | 0.25 | 0.59 | -27.2 |
| November | 1.43 | 0.55 | 2.43 | 1.46 | 0.45 | 2.85 | 0.32 | 0.26 | 0.58 | -39.0 |
| December | 1.46 | 0.75 | 2.30 | 1.75 | 0.98 | 2.66 | NaN | NaN | NaN | 19.3 |
| Average | | | | | | | | | | -25.4 |

To clarify this consideration, we added Table S8 in the revised supplement and modified the following sentence in the revised manuscript:

"Thus, the yearly decline of sulfuric acid is mainly attributed to the decrease of $SO_2$ (by ~25% per year, Table S8)." (Line 202, Page 7)

**REFERENCES**

Chang, X., Zheng, H., Zhao, B., Yan, C., Jiang, Y., Hu, R., Song, S., Dong, Z., Li, S., Li, Z., Zhu, Y., Shi, H., Jiang, Z., Xing, J., and Wang, S.: Drivers of High Concentrations of Secondary Organic Aerosols in Northern China during the COVID-19 Lockdowns, Environmental Science & Technology, 57, 5521-5531, 10.1021/acs.est.2c06914, 2023.

Lu, Y., Yan, C., Fu, Y., Chen, Y., Liu, Y., Yang, G., Wang, Y., Bianchi, F., Chu, B., Zhou, Y., Yin, R., Baalbaki, R., Garmash, O., Deng, C., Wang, W., Liu, Y., Petaja, T., Kerminen, V.-M., Jiang, J., Kulmala, M., and Wang, L.: A proxy for atmospheric daytime gaseous sulfuric acid concentration in urban Beijing, Atmospheric Chemistry and Physics, 19, 1971-1983, 10.5194/acp-19-1971-2019, 2019.

Sarwar, G., Appel, K. W., Carlton, A. G., Mathur, R., Schere, K., Zhang, R., and Majeed, M. A.: Impact of a new condensed toluene mechanism on air quality model predictions in the US, Geosci. Model Dev., 4, 183-193, 10.5194/gmd-4-183-2011, 2011.

Zhao, B., Wang, S., Donahue, N. M., Jathar, S. H., Huang, X., Wu, W., Hao, J., and Robinson, A. L.: Quantifying the effect of organic aerosol aging and intermediate-volatility emissions on regional-scale aerosol pollution in China, Scientific Reports, 6, 28815, 10.1038/srep28815, 2016.

Zhao, B., Zheng, H., Wang, S., Smith, K. R., Lu, X., Aunan, K., Gu, Y., Wang, Y., Ding, D., Xing, J., Fu, X., Yang, X., Liou, K.-N., and Hao, J.: Change in household fuels dominates the decrease in PM2.5 exposure and premature mortality in China in 2005–2015, Proceedings of the National Academy of Sciences, 115, 12401-12406, 10.1073/pnas.1812955115, 2018.

Zheng, H., Cai, S., Wang, S., Zhao, B., Chang, X., and Hao, J.: Development of a unit-based industrial emission inventory in the Beijing–Tianjin–Hebei region and resulting improvement in air quality modeling, Atmos. Chem. Phys., 19, 3447-3462, 10.5194/acp-19-3447-2019, 2019a.

Zheng, H., Zhao, B., Wang, S., Wang, T., Ding, D., Chang, X., Liu, K., Xing, J., Dong, Z., Aunan, K., Liu, T., Wu, X., Zhang, S., and Wu, Y.: Transition in source contributions of PM$_{2.5}$ exposure and associated premature mortality in China during 2005–2015, Environment International, 132, 105111, https://doi.org/10.1016/j.envint.2019.105111, 2019b.

Zheng, H., Chang, X., Wang, S., Li, S., Zhao, B., Dong, Z., Ding, D., Jiang, Y., Huang, G., Huang, C., An, J., Zhou, M., Qiao, L., and Xing, J.: Sources of Organic Aerosol in China from 2005 to 2019: A Modeling Analysis, Environmental Science & Technology, 57, 5957-5966, 10.1021/acs.est.2c08315, 2023.

Zheng, H., Li, S., Jiang, Y., Dong, Z., Yin, D., Zhao, B., Wu, Q., Liu, K., Zhang, S., Wu, Y., Wen, Y., Xing, J., Henneman, L. R. F., Kinney, P. L., Wang, S., and Hao, J.: Unpacking the factors contributing to changes in PM2.5-associated mortality in China from 2013 to 2019, Environment International, 184, 108470, https://doi.org/10.1016/j.envint.2024.108470, 2024.